

# Challenges as enablers for high quality Linked Data: insights from the Semantic Publishing Challenge

Anastasia Dimou[1,2], Sahar Vahdati[3], Angelo Di Iorio[4], Christoph Lange[3,5], Ruben Verborgh[1,2] and Erik Mannens[1,2]

[1] Faculty of Engineering and Architecture, Ghent University, Ghent, Belgium
[2] imec, Leuven, Belgium
[3] Department of Intelligent Systems, University of Bonn, Bonn, Germany
[4] Department of Computer Science and Engineering, University of Bologna, Bologna, Italy
[5] Enterprise Information Systems, Fraunhofer IAIS, Sankt Augustin, Germany

Corresponding author
Anastasia Dimou,
anastasia.dimou@ugent.be

## ABSTRACT

While most challenges organized so far in the Semantic Web domain are focused on comparing tools with respect to different criteria such as their features and competencies, or exploiting semantically enriched data, the Semantic Web Evaluation Challenges series, co-located with the ESWC Semantic Web Conference, aims to compare them based on their output, namely the produced dataset. The Semantic Publishing Challenge is one of these challenges. Its goal is to involve participants in extracting data from heterogeneous sources on scholarly publications, and producing Linked Data that can be exploited by the community itself. This paper reviews lessons learned from both (i) the overall organization of the Semantic Publishing Challenge, regarding the definition of the tasks, building the input dataset and forming the evaluation, and (ii) the results produced by the participants, regarding the proposed approaches, the used tools, the preferred vocabularies and the results produced in the three editions of 2014, 2015 and 2016. We compared these lessons to other Semantic Web Evaluation Challenges. In this paper, we (i) distill best practices for organizing such challenges that could be applied to similar events, and (ii) report observations on Linked Data publishing derived from the submitted solutions. We conclude that higher quality may be achieved when Linked Data is produced as a result of a challenge, because the competition becomes an incentive, while solutions become better with respect to Linked Data publishing best practices when they are evaluated against the rules of the challenge.

## INTRODUCTION

The Semantic Web aims to extend the human-readable Web by encoding the semantics of resources in a machine-comprehensible and reusable fashion. Over the past years, a growing amount of research on publishing and consuming Linked Data, i.e., data represented and made available in a way that maximizes reusability, has facilitated Semantic Web adoption. However, one of the remaining issues is lack of high quality Linked

Data. A promising means to foster and accelerate the publication of such high quality Linked Data is the organization of *challenges*: competitions during which participants complete tasks with innovative solutions that are then ranked in an objective way to determine the winner. A significant number of challenges has been organized so far, including the Semantic Web Challenge (see http://challenge.semanticweb.org/), its Big Data Track formerly known as the Billion Triples Challenge, and the LinkedUp Challenge (http://linkedup-challenge.org/), to mention a few of the longest lasting. However, these challenges targeted broad application domains and were more focused on innovative ways of exploiting Semantic Web enabled tools (*Linked Data consumption*) than on the output actually produced (*Linked Data production*). Therefore, such challenges enable advancement of Semantic Web technology but overlook the possibility of also advancing Linked Datasets per se.

This paper focuses on a series of Challenges in the Semantic Publishing domain. *Semantic publishing* is defined as "the enhancement of scholarly publications by the use of modern Web standards to improve interactivity, openness and usability, including the use of ontologies to encode rich semantics in the form of machine-readable RDF metadata" by *Shotton (2009)*. The 2014 Semantic Publishing Challenge, was themed "Assessing the Quality of Scientific Output" (*Lange & Di Iorio, 2014*) (2014 SemPub Challenge, http://2014.eswc-conferences.org/semantic-publishing-challenge.html), in 2015 we mentioned the techniques more explicitly by appending "... by Information Extraction and Interlinking" (*Di Iorio et al., 2015*) (2015 SemPub Challenge, http://2015.eswc-conferences.org/important-dates/call-SemPub), and in 2016 we generalized to "... in its Ecosystem" to emphasize the multiple dimensions of scientific quality and the potential impact of producing Linked Data about it (*Dimou et al., 2016*) (2016 SemPub Challenge, http://2016.eswc-conferences.org/assessing-quality-scientific-output-its-ecosystem).

According to *Miller & Mork (2013)*, extracting, annotating and sharing scientific data (by which, here, we mean standalone research datasets, data inside documents, as well as metadata about datasets and documents) and then building new research efforts on them, can lead to a data value chain producing value for the scholar and Semantic Web community. On the one hand, the scholar community benefits from a challenge that produces data, as the challenge results in more data and in data of higher quality being available to the community to exploit. On the other hand, the Semantic Web community benefits: participants optimize their tools towards performance in this particular challenge, but such optimisations may also improve the tools in general. Once such tools are reused, any other dataset benefits from their advancements, because the processes producing them has been improved. However, bootstrapping and enabling such value chains is not easy.

In a recent publication (*Vahdati et al., 2016*), we discussed lessons we learned from our experience in organizing the first two editions of the Semantic Publishing Challenge—mainly from the perspective of how to improve the organization of further editions and of providing a better service to the scholar community. The lessons are related to the challenge organization, namely defining the tasks, building the input datasets and performing the

evaluation, as well as lessons we learned by studying the solutions, with respect to the methodologies, tools and ontologies used, and data produced by the participants. We organized the third edition based on these lessons learned.

In this paper, we revise our lessons learned, taking into consideration experience gained by organizing the challenge's third edition, whose results validate in principle our lessons learned. We argue that challenges may act as enablers for the generation of higher quality Linked Data, because of the competitive aspect. However, organizing a successful challenge is not an easy task. Therefore, the goal of this paper is to distill *generic best practices*, which could be applied to similar events, rendering the challenge tasks into meaningful milestones for efficient Linked Data generation and publishing. To achieve that, we validated the generalizability of our lessons learned against the other Semantic Web Evaluation Challenges (2014 Semantic Web Evaluation Challenges, http://2014.eswc-conferences.org/important-dates/call-challenges.html, 2015 Semantic Web Evaluation Challenges, http://2015.eswc-conferences.org/call-challenges, 2016 Semantic Web Evaluation Challenges, http://2016.eswc-conferences.org/call-challenges).

We concluded that our lessons learned are applicable to other challenges too; thus they can be considered *best practices* for organizing a challenge. Other challenge organizers may benefit from relying on these best practices when organizing their own challenge. Additionally, we thoroughly analyze and report best practices followed by the Linked Data that the *solutions* to our challenge's tasks produce. Our study of the different solutions provides insights regarding different approaches that address the same task, namely it acts as if the challenge benchmarks those different solutions against a common problem. Last, we assess based on the produced datasets how the challenge organization reinforces increasing Linked Data quality in respect to the different Linked Data dimensions identified by *Zaveri et al. (2016)*.

Thus, besides the scholarly community and the CEUR-WS.org open access repository, which is the owner of the underlying data, the broader Linked Data community may benefit from looking into our cumulative results. Other Linked Data owners may find details on different approaches dealing with the same problem and the corresponding results they produce. Taking them into consideration, they can determine their own approach for an equivalent case or even consider launching a corresponding challenge to determine the best performing tool with respect to the desired results and consider this one for their regular long term use. Moreover, other Linked Data publishers may advise the results or consider the best practices as their guidelines for improving their tools and thus their results.

In summary, our contributions are:

- an *outline of challenges* organized in the field of Linked Data and Semantic Web technologies,
- an *exhaustive analysis* of all solutions to every task of all editions of the Semantic Publishing Challenge series,
- a *systematic discussion of lessons* that we have learned from organizing the Semantic Publishing Challenge, and
- a structured set of *best practices* for organizing similar challenges, resulting from validating our lessons against other Semantic Web Evaluation Challenges.

The remainder of the paper is structured as follows: 'Background and Related Work' section reviews related work; in particular it sets the background for our study by recapitulating the Semantic Publishing Challenges run so far and comparing them to related challenges. 'Best Practices for Challenge Organization' section revisits the lessons learned, taking into consideration all three editions, validates them against other challenges and concludes in best practices for organizing such challenges. 'Challenge Solutions Analysis' section exhaustively and cumulatively analyses the solutions submitted to all tasks of all challenges in the series. 'Discussion: Challenge Impact on Linked Data Quality' section reviews the Semantic Publishing Challenges as a means of assessing the quality of data, and 'Conclusions' section summarizes our conclusions.

# BACKGROUND AND RELATED WORK

This section sets the background of the Semantic Publishing Challenges so far. 'State of the art on previously organized challenges' section summarizes other challenges, mainly those run in the Semantic Web community. Then, 'Semantic Publishing Challenge: 2014–2016' section recapitulates the Semantic Publishing Challenges run so far, including the definitions of their tasks, and their outcomes.

## State of the art on previously organized challenges

Several related challenges were organized in the past for different purposes and application domains. In this section, we summarize the most well-known, long-lasting and closely related challenges in the Semantic Web field. Where applicable, we report on systematic *reviews* of challenges for lessons learned.

### Ontology matching challenges

The *Ontology Matching Challenges* (http://ontologymatching.org/) have been organized since 2004 by the Ontology Alignment Evaluation Initiative (OAEI, http://oaei. ontologymatching.org/) and co-located with several top Information Systems and Web conferences such as WWW (World Wide Web Conferences, https://en.wikipedia.org/ wiki/International_World_Wide_Web_Conference) or VLDB (Very Large Databases Conferences, https://en.wikipedia.org/wiki/VLDB). It aims to forge a consensus for evaluating the different emerging methods for schema or ontology matching. The OAEI aims to assess the strengths and weaknesses of alignment/matching systems, compare the performance of techniques, and improve evaluation techniques to help improving the work on ontology alignment/matching through evaluating the techniques' performances. Following a similar structure as the Semantic Publishing Challenge, the OAEI challenge provides a list of test ontologies as training datasets. The SEALS infrastructure (http://oaei.ontologymatching.org/2016/seals-eval.html) to evaluate the results has been made available since 2011. The results are presented during the Ontology Matching workshop, which is usually co-located with the International Semantic Web Conference (ISWC, http://swsa.semanticweb.org/content/international-semantic-web-conference-iswc). The tests and results of the challenge are published for further analysis.

### Semantic Web Challenge

The *Semantic Web Challenge* (http://challenge.semanticweb.org/) aims to apply Semantic Web techniques in building online end-user applications that integrate, combine and deduce information needed to assist users in performing tasks. It features a track about Big Data designed to demonstrate approaches which can work on Web scale using realistic Web-quality data. The Big Data Track, formerly known as the *Billion Triples Challenge (BTC)*, started from 2008 mostly co-located with ISWC. The Billion Triples Challenge aimed to demonstrate the capability of Semantic Web technologies to process very large and messy data as typically found on the Web. The track was renamed to "Big Data Track" because very large data sets are now ubiquitous and the competition was opened to broader range of researchers dealing with their own big data. The functionality of submitted solutions is open but, to address real scalability issues, it forces all participants to use a specific Billion Triple Challenge Dataset provided by the challenge's organizers.

### Question Answering over Linked Data (QALD)

The *Question Answering over Linked Data (QALD)* challenge (http://qald.sebastianwalter.org/) (*Lopez et al., 2013*; *Unger et al., 2015*) focuses on answering natural language or keyword-based questions over linked datasets. Co-located with the ESWC Semantic Web Conference (ESWC, http://eswc-conferences.org/) in its first two editions in 2011 and 2013, it moved to the Conference and Labs of the Evaluation Forum (CLEF, https://en.wikipedia.org/wiki/Conference_and_Labs_of_the_Evaluation_Forum) for the three following editions, to return to ESWC as a part of its Semantic Web Evaluation Challenges track explained below. In all editions, a set of up to 340 questions over DBpedia (https://dbpedia.org) served as input; participants were expected to answer these questions. The 2013–2016 editions had a task on multilingual questions, while from 2014, a task on hybrid question answering over RDF and free text was added. Some editions considered alternative datasets, e.g., about drugs or music, and had alternative sub-tasks on answering questions over interlinked datasets or finding lexicalizations of ontological terms. Only few submitted solutions address the question/answering issues over a distributed and large collection of interconnected datasets.

The first two editions of the QALD Challenge were reviewed (*Lopez et al., 2013*); similarly to our work, this review "discuss[es] how the second evaluation addressed some of the issues and limitations which arose from the first one, as well as the open issues to be addressed in future competitions". Like us, Lopez et al. present the definition of the QALD challenge's tasks and the datasets used, and draw conclusions for the subsequent evaluation of question answering systems from reviewing concrete results of the first two challenge editions. Their review of related work includes a review of methods for evaluating question answering systems, whereas the Semantic Publishing Challenge was created to address the lack of such methods for evaluating semantic publishing tools (cf. 'Semantic Publishing Challenge: 2014–2016'). We additionally present lessons learned for challenge organization ('Best Practices for Challenge Organization') and about semantic publishing tools ('Challenge Solutions Analysis'), which, together, constitute the main contribution of this paper.

### LAK Challenges

The *Learning Analytics and Knowledge Challenges* (LAK Challenges; see http://meco.l3s.uni-hannover.de:9080/wp2/?page_id=18) use a specific dataset of structured metadata from research publications in the field of learning analytics. The challenge was organized in 2011 for the first time and has so far continued yearly with the LAK conference. Beyond merely publishing the data, the LAK challenges encourage its innovative use and exploitation. Participants submit a meaningful use case of the dataset in the scope of six topic categories, such as comparison of the LAK and EDM (Educational Data Mining) communities, innovative applications to explore, navigate and visualize, enrichment of the Dataset, and usage of the dataset in recommender systems. Considering that a lot of information is still available only in textual form, the submitted approaches can not only deal with the specific character of structured data. The aim for further challenges is to combine solutions for processing both structured and unstructured information from distributed datasets.

### LinkedUp

The *LinkedUp* challenge was run by the LinkedUp project (Linking Data for Education, http://linkedup-project.eu/) since 2014. The main purpose of the project was to push educational organizations to make their data publicly available on the Web. One of the activities towards this purpose was to organize the *LinkedUp Challenge*. The three editions of the challenge focused on three different levels of maturity: demo prototypes and applications, innovative tools and applications, and mature data-driven applications. Participants were asked to submit demos of tools that analyze and/or integrate open Web data for educational purposes. For all the above challenges, the participants were asked to submit a scientific paper along with their tool and dataset.

*D'Aquin et al. (2014)* present lessons learned from the LinkedUp project (Linking Web Data for Education). However, their paper provides a summary of the outcomes of the project, including a summary of the LinkedUp Challenge, rather than a systematically structured account of lessons learned.

### Dialog State Tracking Challenge (DSTC)

The challenge series review that is most closely related to ours in its methodology has been carried out by *Williams, Raux & Henderson (2016)* over a challenge series from a field of computer science that is related to semantics but not to the Web: the Dialog State Tracking Challenge (DSTC, http://workshop.colips.org/dstc5/) on "correctly inferring the state of [a] conversation [...] given all of the dialog history". Like our review, the one of DSTC is based on three editions of a challenge, each of which built on its predecessor's results, and it presents the definition of the challenge's tasks and the datasets used. Like we do in 'Challenge Solutions Analysis' section, they provide a structured overview of the submissions to the DSTC challenges. However, the focus of their review is on the evolution of tools in their domain of dialog state tracking, whereas our review additionally covers lessons learned for challenge design (cf. 'Best Practices for Challenge Organization'), besides tools in the domain of Semantic publishing.

**Table 1** Semantic Web Evaluation Challenges.

| Abbreviation | Challenge | Years |
| --- | --- | --- |
| SemPub | Semantic Publishing Challenge | 2014, 2015, 2016 |
| CLSA | (Concept-Level) Sentiment Analysis Challenge | 2014, 2015, 2016 |
| RecSys | Linked Open Data-Enabled Recommender System Challenge | 2014, 2015 |
| OKE | Open Knowledge Extraction Challenge | 2015, 2016 |
| SAQ | Schema-agnostic Queries over Linked Data | 2015 |
| QALD | Open Challenge on Question Answering over Linked Data | 2016 |
| Top-K | Top-K Shortest Path in Large Typed RDF Graphs Challenge | 2016 |

### Other related works

There are further related works and challenges that we consider out of the scope, as they are not focused on Linked Data sets. For example, the *AI Mashup Challenge* (http://aimashup.org/) as a part of the ESWC conference focused on innovative mashups, i.e., web applications combining multiple services and datasets, that were evaluated by a jury. Information Retrieval campaigns are a series of comparative evaluation methods that originate from the 1960s and are used to compare various retrieval strategies or systems. As an example of such campaigns SemEval (Semantic Evaluation) (SemEval campaigns, http://alt.qcri.org/semeval2016/) is one of the ongoing series of evaluations of computational semantic analysis systems with a focus on Textual Similarity and Question Answering and Sentiment Analysis (*Clough & Sanderson (2013)*). The *Computational Linguistics Scientific Document Summarization Shared Task (CL-SciSumm)* (http://wing.comp.nus.edu.sg/cl-scisumm2016/) is based on a corpus of annotated documents; tasks focus on correctly identifying the underlying text that a summary refers to, but also on generating summaries.

### Semantic Web Evaluation Challenges

The *Semantic Web Evaluation Challenges*, including our Semantic Publishing Challenge, aim at developing a set of common benchmarks and establish evaluation procedures, tasks and datasets in the Semantic Web field. They are organized as an official track of the ESWC Semantic Web Conference, which introduces common standards for its challenges, e.g., common deadlines for publishing the training and evaluation datasets. The purpose of the challenges is to showcase methods and tools on tasks common to the Semantic Web and adjacent disciplines, in a controlled setting involving rigorous evaluation. Each Semantic Web Evaluation Challenge is briefly described here and all of them are summarized at Table 1.

*Concept-Level Sentiment Analysis Challenge.* The *Concept-Level Sentiment Analysis Challenge (CLSA)* focuses on semantics as a key factor for detecting the sentiment of a text, rather than just performing a lexical analysis of text; cf. *Reforgiato Recupero & Cambria (2014)* and *Reforgiato Recupero, Dragoni & Presutti (2015)*. Participants are asked to use Semantic Web technology to improve their sentiment analysis system and to measure the performance of the system (http://alt.qcri.org/semeval2015/task12/) within the Sentiment

Analysis track of the SEMEVAL 2015 workshop, (http://alt.qcri.org/semeval2015/). An automatic evaluation tool (ESWC-CLSA 2015, https://github.com/diegoref/ESWC-CLSA) was applied to the submissions; it was made available to the participants before their submission. In the second edition, participants were asked to submit a concept-level sentiment analysis engine that exploited linked datasets such as DBpedia.

*Linked Open Data-Enabled Recommender Systems Challenge.* The *Linked Open Data-Enabled Recommender Systems Challenge* (*Di Noia, Cantador & Ostuni, 2014*) was designed with two main goals: (i) establish links between the two communities of recommender systems and Semantic Web, and (ii) develop content-based recommendation systems using interlinking and other Semantic Web and Technologies. The first edition featured three independent tasks related to a book recommendation use case. While the first edition was successful, the second edition was canceled because it had no participants.

*Open Knowledge Extraction Challenge.* The *Open Knowledge Extraction Challenge (OKE)* focuses on content extraction from textual data using Linked Data technology (*Nuzzolese et al., 2015*). The challenge was divided into two sub-tasks (OKE Challenge 2016, https://github.com/anuzzolese/oke-challenge-2016#tasks-overview) focusing on entity recognition and entity typing. The participants of the challenge were the developers of four different well-known systems in this community. The three defined tasks were focused on (a) entity recognition, linking and typing for knowledge base population, (b) entity typing for vocabulary and knowledge base enrichment and (c) Web-scale knowledge extraction by exploiting structured annotation. The submissions were evaluated using two different methods: (i) using datasets for training purposes and for evaluating the performance of the submitted approaches, and (ii) establishing an evaluation framework to measure the accuracy of the systems. The applications of task 1 and 2 were published as web services with input/output provided in the NLP Interchange Format NIF (http://persistence.uni-leipzig.org/nlp2rdf/).

*Schema-Agnostic Queries over Linked Data Challenge.* The *Schema-Agnostic Queries over Linked Data Challenge (SAQ)* was designed to invite schema-agnostic query approaches and systems (*Freitas & Unger, 2015*). The goal of this challenge is to improve querying approaches over complex databases with large schemata and to relieve users from the need to understand the database schema. Tasks were defined for two types of queries: schema-agnostic SPARQL queries and schema-agnostic keyword-based queries. Participants were asked to submit the results together with their approach without changing the query syntax but with different vocabularies and structural changes. A gold standard dataset was used to measure precision, recall and F1-score.

## Semantic Publishing Challenge: 2014–2016

In this section, we briefly summarize the history of the Semantic Publishing Challenge to provide the necessary background for the following discussion. More detailed reports for each edition have been published separately by *Lange & Di Iorio (2014)*, *Di Iorio et al. (2015)* and *Dimou et al. (2016)*.

We sought a way to challenge the semantic publishing community to accomplish tasks whose results could be compared *in an objective way*. After some preliminary discussion, we focused on *information extraction tasks*. The basic idea was to provide as input some scholarly papers—in multiple formats—and some queries in natural language. Participants were asked to extract data from these papers and to publish them as an RDF dataset that could be used to answer the input queries. The best performing approach was identified automatically by comparing the output of the queries in the produced datasets against a gold standard, and by measuring precision and recall. Our selection of queries was motivated by quality assessment scenarios complementary to the traditional metrics based on counting citations: how can the extracted information serve as indicators for the quality of scientific output such as publications or events. The same motivation, structure and evaluation procedure have been maintained in the following years, with some improvements and  extensions.

All challenge's series' tasks (see 'Tasks evolution' section), the input to the tasks, namely the training and evaluation datasets (see 'Input: training and evaluation datasets' section), the output, namely the submitted solutions and the produced dataset (see 'Output: solutions and datasets produced' section) and how their evaluation was conducted (see 'Tasks evaluation' section) are briefly explained below.

### Tasks evolution

Table 2 summarizes the tasks' full history. For each year and each task, we highlight the data source and the format of the input files, along with a short description of the task and a summary on the participation.

*2014 edition tasks.* The first edition had two main tasks (Task 1 and Task 2) and an open task (Task 3; see *Lange & Di Iorio (2014)* for full details and statistics of this challenge's edition).

For Task 1, the participants were asked to extract information from selected CEUR-WS.org workshop proceedings volumes to enable the computation of indicators for the workshops' quality assessment. The input files were HTML tables of content using different levels of semantic markup, as well as PDF full text. The participants were asked to answer twenty queries. For Task 2, the input dataset included XML-encoded research papers, derived from the PubMedCentral and Pensoft Open Access archives. The participants were asked to extract data about citations to assess the value of articles, for instance by considering citations' position in the paper, their co-location with other citations, or their purpose. In total, they were asked to answer ten queries. Dataset and queries were completely disjoint from Task 1.

After circulating the call for submissions, we received feedback from the community that mere information extraction, even if motivated by a quality assessment use case, was not the most exciting task related to the future of scholarly publishing, as it assumed a traditional publishing model. Therefore, to address the challenge's primary target, i.e., 'publishing' rather than just 'metadata extraction', we widened the scope by adding an open task (Task 3). Participants were asked to showcase data-driven applications that would eventually support publishing. We received a good number of submissions; winners were selected by a jury.

**Table 2  Semantic Publishing Challenge evolution from 2014 to 2016.**

| | | 2014 edition | 2015 edition | 2016 edition |
|---|---|---|---|---|
| **Task 1** | **Task** | Extracting data on workshops history and participants | Extracting data on workshops history and participants | Extracting data on workshops history and participants |
| | **Source** | CEUR-WS.org proceedings volumes | CEUR-WS.org proceedings volumes | CEUR-WS.org proceedings volumes |
| | **Format** | HTML and PDF | HTML | HTML |
| | **Solutions** | 3 | 4 | 0 |
| | **Awards** | best performance innovation | best performance innovation | – |
| | **Decision** | – | chairs' assessment | chairs' assessment |
| **Task 2** | **Task** | Extracting data on citations | Extracting data on citations, affiliations, fundings | Extracting data on internal structure, affiliations, fundings |
| | **Source** | PubMed | CEUR-WS.org | CEUR-WS.org |
| | **Format** | XML | PDF | PDF |
| | **Solutions** | 1 | 6 | 5 |
| | **Awards** | – | best performance most innovative | best performance most innovative |
| | **Decision** | – | chairs' assessment | chairs' assessment |
| **Task 3** | **Task** | Open task: showcasing semantic publishing applications | Interlinking cross-dataset entities | Interlinking cross-dataset entities cross-task entities |
| | **Source** | – | CEUR-WS.org, Colinda DBLP, Springer LD Lancet, SWDF | CEUR-WS.org, Colinda DBLP, Springer LD |
| | **Format** | – | RDF | RDF |
| | **Solutions** | 4 | 0 | 0 |
| | **Awards** | Most innovative (jury assessment) | – | – |

*2015 edition tasks.* In 2015 we were asked to include only tasks that could be evaluated in a fully objective manner, and thus we discarded the 2014's edition open task (Task 3).

While Task 1 queries remained largely stable from 2014 to 2015, the queries for Task 2 changed. We transformed Task 2 into a PDF mining task, instead of XML, and thus moved all PDF-related queries there. The rationale was to differentiate tasks on the basis of the competencies and tools required to solve them. Since the input format was completely new and we expected different teams to participate (as actually happened), we wanted to explore new areas and potentially interesting information. In fact, we asked participants to extract data not only on citations but also on affiliations and fundings. The number of queries remained unchanged (ten in total). We also decided to use the same data source for both tasks, and to make them interplay. CEUR-WS.org data has become the central focus of the whole challenge, for two reasons: on the one hand, the data provider (CEUR-WS.org) takes advantage of a broader community that builds on its data, which, before the Semantic Publishing Challenges, had not been available as Linked Data[1]. On the other hand, data consumers gain the opportunity to assess the quality of scientific venues by taking a deeper look into their history, as well as the quality of the publications.

In 2015, we also introduced a new Task 3. Instead of being an open task, Task 3 was focused on interlinking the dataset produced by the winners of Task 1 from the 2014 edition of the Semantic Publishing Challenge with related datasets in the Linked Data Cloud.

[1]On a more pragmatic level, a further reason was that one of the challenge organizers, Christoph Lange, has been technical editor of CEUR-WS.org since 2013 and thus has (i) the mandate to advance this publication service technically, and (ii) a deep understanding of the data.

*2016 edition tasks.* The tasks of the 2016 edition were designed to ensure continuity and to allow previous participants to use and refine their tools.

In particular, Task 1 was unchanged except for some minor details on queries. Task 2 was still on PDF information extraction but queries were slightly changed: considering the interest and results of the participants in the past, we did not include citations any more. Rather, we added some queries on the identification of the structural components of the papers (table of contents, captions, figures and tables) and maintained queries on funding agencies and projects. In total, we had ten queries in 2016 as well.

Task 3 remained the same but it was repurposed. Instead of only aiming for cross-dataset links between the dataset produced by the Task 1 winners of the previous edition of the challenge and other, external datasets, Task 3 now focused on interlinking the datasets produced by the winners of Task 1 and Task 2 of the 2015 edition. Thus, the task aimed not only at *cross-dataset* but also at *cross-task* links: the goal was to link entities identified in the CEUR-WS.org website with the same entities that were extracted from the proceedings papers. Moreover, the number of external datasets was reduced.

### Input: training and evaluation datasets

In this section we give an overview of the datasets used for the above mentioned tasks. These datasets were incrementally refined and, as discussed below in 'Dataset continuity', some valuable indications can be taken from their analysis. For each task, and for each year, we published two datasets: (i) a training dataset (TD) on which the participants could test and train their extraction tools and (ii) an evaluation dataset (ED) made available a few days before the final submission and used as input for the final evaluation.

*Training and evaluation dataset for task 1.* The CEUR-WS.org workshop proceedings volumes served as the source for selecting the training and evaluation datasets of Task 1 in all challenge editions. In this data source, which included data spanning over 20 years, workshop proceedings volumes were represented in different formats and at different levels of encoding quality and semantics. An HTML4 main index page (CEUR-WS, http://ceur-ws.org/) links to all workshop proceedings volumes, which have HTML tables of contents and contain PDF or PostScript full texts. A mixture of different HTML formats (no semantic markup at all, different versions of microformats, RDFa) were chosen for both the training and evaluation datasets. The training dataset comprised all volumes of several workshop series, including, e.g., the Linked Data on the Web workshop at the WWW conference, and all workshops of some conferences, e.g., of several editions of ESWC. In 2014 and 2015, the evaluation dataset was created by adding further workshops on top of the training dataset. To support the evolution of extraction tools, the training datasets of 2015 and 2016 were based on the unions of the training and evaluation datasets of the previous years. In 2015 and 2016, the Task 1 dataset of the previous year served as an input to Task 3.

*Training and evaluation dataset for task 2.* In 2014, the datasets for Task 2 included XML files encoded in JATS, (http://jats.nlm.nih.gov/) and TaxPub, (https://github.com/plazi/TaxPub), an official extension of JATS customized for taxonomic treatments (*Catapano,*

*2010*). The training dataset consisted of 150 files from 15 journals, while the evaluation dataset included 400 papers and was a superset of the training dataset. In 2015, we switched to PDF information extraction: the training dataset included 100 papers taken from some of the workshops analyzed in Task 1, while the evaluation dataset included 200 papers from randomly selected workshops (uniform to the training dataset). In 2016, we reduced the number of papers increasing the cases for each query. Thus, we included 50 PDF papers in the training and 40 in the evaluation dataset. Again, the papers were distributed in the same way and used different styles for headers, acknowledgments and structural components.

*Training and evaluation dataset for task 3.* The training dataset for Task 3 consists of the CEUR-WS.org dataset produced by the 2014 winning tool of Task 1 (2014 CEUR-WS dataset, https://github.com/ceurws/lod/blob/master/data/ceur-ws.ttl), COLINDA (http://www.colinda.org/), DBLP (http://dblp.l3s.de/dblp++.php), Lancet (http://www.semanticlancet.eu/), SWDF (http://data.semanticweb.org/), and Springer LD (http://lod.springer.com/) in 2015 and the CEUR-WS.org datasets produced by the 2015 winning tools of Task 1 (2015 CEUR-WS Task 1 dataset, http://rml.io/data/SPC2016/CEUR-WS/CEUR-WStask1.rdf.gz) and Task 2 (2015 CEUR-WS Task 2 dataset, http://rml.io/data/SPC2016/CEUR-WS/CEUR-WStask2.rdf.gz), of COLINDA, DBLP, and Springer LD in 2016.

### Output: solutions and datasets produced

There were four distinct solutions in total for Task 1 during the three editions of the challenge, eight distinct solutions in total for Task 2 and none for Task 3 during the last two editions. All solutions for each task are briefly summarized here.

*Task 1.* There were four distinct solutions proposed to address Task 1 in 2014 and 2015 editions of the challenge. Three participated in both editions, whereas the fourth solution participated only in 2015. All solutions are briefly introduced here and summarized in Tables 3–7. Table 3 provides details about the methodologies, approach and implementation each solution followed. Table 4 summarizes the model and vocabularies/ontologies each solution used (both for Task 1 and Task 2), whereas Table 7 provides statistics regarding the dataset schema/entities and triples/size each solution produced (again both for Task 1 and Task 2). Last, Table 5 summarizes the data model each solution considered and Table 6 the number of instances extracted and annotated per concept for each solution.

*Solution 1.1. Kolchin et al. (2015)* and *Kolchin & Kozlov (2014)* presented a case-specific crawling based approach for addressing Task 1. It relies on an extensible template-dependent crawler that uses sets of special predefined templates based on XPath and regular expressions to extract the content from HTML and convert it in RDF. The RDF is then processed to merge resources using fuzzy-matching. The use of the crawler turns the system tolerant to invalid HTML pages. This solution improved its precision in 2015 as well the richness of the data model.

**Table 3  Task 1 solutions: their primary analysis methods, methodologies, implementations basis and evaluation results.**

|  | Solution 1.1 | Solution 1.2 | Solution 1.3 | Solution 1.4 |
|---|---|---|---|---|
| **Publications** | *Kolchin et al. (2015)* <br> *Kolchin & Kozlov (2014)* | *Heyvaert et al. (2015)* <br> *Dimou et al. (2014)* | *Ronzano et al. (2015)* <br> *Ronzano, Del Bosque & Saggion (2014)* | *Milicka & Burget (2015)* <br> – |
| **Primary analysis** |  |  |  |  |
| structure-based |  | ✓ | ✓ |  |
| syntactic-based | ✓ |  |  | ✓ |
| linguistic-based |  |  | ✓ |  |
| layout-based |  |  |  | ✓ |
| **Methodology** |  |  |  |  |
| method | Crawling | Generic solution for abstracted mappings | Linguistic and structural analysis | Visual layout multi-aspect content analysis |
| case-specific | ✓ |  | ✓ (partly) | ✓ (partly) |
| template-based | ✓ | ✓ |  |  |
| NLP/NER |  |  | ✓ | ✓ |
| **Implementation** |  |  |  |  |
| basis | n/a | RML | GATE | FITLayout |
| language | Python | Java | Java | Java, HTML |
| rules language | XPath | RML, CSS | JAPE | HTML,CSS |
| code/rule separation |  | ✓ | ✓ |  |
| regular expressions | ✓ | ✓ | ✓ | ✓ |
| external services |  |  | ✓ | ✓ |
| open source | ✓ | ✓ |  | ✓ |
| license | MIT | MIT | – | GPL-3.0 |
| **Evaluation** |  |  |  |  |
| precision improvement | 11.1% | 11.4% | 10.7% | – |
| recall improvement | 11.3% | 11.3% | 10.9% | – |
| best performing | ✓ (2014) |  |  | ✓ (2015) |
| most innovative |  |  | ✓ (2014) | ✓ (2015) |

*Solution 1.2. Heyvaert et al. (2015)* and *Dimou et al. (2014)* exploited a generic tool for generating RDF data from heterogeneous data. It uses the RDF Mapping Language (RML http://rml.io) to define how data extracted from CEUR-WS.org Web pages should be semantically annotated. RML extends R2RML (https://www.w3.org/TR/r2rml/) to express mapping rules from heterogeneous data to RDF. CSS3 selectors (CSS3, https://www.w3.org/TR/selectors/) are considered to extract the data from the HTML pages. The RML mapping rules are parsed and executed by the RML Processor (https://github.com/RMLio/RML-Mapper). In 2015 the solution reconsidered its data model and was extended to validate both the mapping documents and the final RDF, resulting in an overall improved quality dataset.

*Solution 1.3. Ronzano et al. (2015)* and *Ronzano, Del Bosque & Saggion (2014)* designed a case-specific solution that relies on chunk-based and sentence-based Support Vector Machine (SVM) classifiers which are exploited to semantically characterize parts of

**Table 4  Task 1 and 2 solutions: the vocabularies used to annotate the data.**

|  | Sol 1.1 | Sol 1.2 | Sol 1.3 | Sol 1.4 | Sol 2.1 | Sol 2.2 | Sol 2.3 | Sol 2.4 | Sol 2.5 | Sol 2.6 | Sol 2.7 | Sol 2.8 |
|---|---|---|---|---|---|---|---|---|---|---|---|---|
| **bibo**[1] | ✓ | ✓ |  | ✓ |  |  |  | ✓ | ✓ |  |  |  |
| **co**[2] |  |  | ✓ |  |  |  |  |  |  | ✓ |  |  |
| **DBO**[3] | ✓ |  | ✓ | ✓ |  | ✓ |  |  | ✓ |  |  |  |
| **DC**[4] | ✓ | ✓ | ✓ | ✓ | ✓ |  |  |  | ✓ |  |  | ✓ |
| **DCterms**[5] | ✓ |  |  | ✓ | ✓ |  |  | ✓ |  | ✓ |  |  |
| **event**[6] |  | ✓ |  |  |  |  |  |  | ✓ |  |  |  |
| **FOAF**[7] | ✓ | ✓ | ✓ | ✓ |  |  |  | ✓ | ✓ | ✓ |  | ✓ |
| **schema**[8] |  |  |  |  |  |  | ✓ | ✓ |  |  |  |  |
| **SKOS**[9] | ✓ |  |  |  |  |  |  |  |  |  |  |  |
| **SPAR**[10] |  | ✓ | ✓ |  |  |  | ✓ | ✓ |  | ✓ |  | ✓ |
| **BiRO** |  |  | ✓ |  |  |  |  |  |  | ✓ |  |  |
| **CiTO** |  |  |  |  |  |  |  |  |  |  |  | ✓ |
| **DoCO** |  |  |  |  |  |  | ✓ | ✓ |  |  |  | ✓ |
| **FaBiO** |  | ✓ | ✓ |  |  |  | ✓ | ✓ |  | ✓ |  |  |
| **FRAPO** |  |  |  |  |  |  | ✓ | ✓ |  |  |  |  |
| **FRBR** |  |  | ✓ |  |  |  |  |  |  |  |  |  |
| **PRO** |  |  | ✓ |  |  |  | ✓ | ✓ |  | ✓ |  |  |
| **SWC**[11] | ✓ |  |  | ✓ |  |  |  | ✓ |  |  |  |  |
| **SWRC**[12] | ✓ | ✓ | ✓ | ✓ |  |  |  |  | ✓ | ✓ |  |  |
| **timeline**[13] | ✓ |  |  | ✓ |  |  |  |  |  |  |  |  |
| **vcard**[14] |  |  | ✓ | ✓ | ✓ |  |  |  |  |  |  |  |
| **custom** |  |  |  |  |  |  |  | ✓ | ✓ |  | ✓ | ✓ |

**Notes.**
[1] bibo, http://purl.org/ontology/bibo/.
[2] Collections Ontology, http://purl.org/co/.
[3] DBO, http://dbpedia.org/ontology/.
[4] DC, http://purl.org/dc/elements/1.1/.
[5] DCTerms, http://purl.org/dc/terms/.
[6] event ontology, http://purl.org/NET/c4dm/event.owl#.
[7] FOAF, http://xmlns.com/foaf/0.1/.
[8] Schema.org, http://schema.org.
[9] SKOS, http://www.w3.org/2004/02/skos/core#.
[10] SPAR, http://www.sparontologies.net/.
[11] SWC, http://data.semanticweb.org/ns/swc/ontology#.
[12] SWRC, http://swrc.ontoware.org/ontology#.
[13] timeline ontology, http://purl.org/NET/c4dm/timeline.owl#.
[14] VCard, http://www.w3.org/2006/vcard/ns#.

CEUR-WS.org proceedings textual contents. Thanks to a pipeline of text analysis components based on the GATE Text Engineering Framework (GATE, https://gate.ac.uk/), each HTML page is characterized by structural and linguistic features: these features are then exploited to train the classifiers on the ground-truth provided by the subset of CEUR-WS.org proceedings with microformat annotations. A heuristic-based annotation sanitizer is applied to fix classifiers imperfections and interlink annotations. The produced dataset is also extended with information retrieved from external resources.

*Solution 1.4. Milicka & Burget (2015)* presented an application of the FITLayout framework (http://www.fit.vutbr.cz/~burgetr/FITLayout/). This solution participated

Dimou et al. (2017), *PeerJ Comput. Sci.*, DOI 10.7717/peerj-cs.105

**Table 5** Statistics about the model (Task 1—2014 and 2015 editions).

| Year | Solution 1.1 | | Solution 1.2 | | Solution 1.3 | | Solution 1.4 |
| --- | --- | --- | --- | --- | --- | --- | --- |
| | 2014 | 2015 | 2014 | 2015 | 2014 | 2015 | 2015 |
| **Conferences** | swc:OrganizedEvent | swc:OrganizedEvent | swc:Event | bibo:Conference | swrc:Event | swrc:Conference | swrc:ConferenceEvent |
| **Workshops** | bibo:Workshop | bibo:Workshop | swc:Event | bibo:Workshop | swrc:Event | swrc:Workshop | swrc:Section |
| **Proceedings** | swrc:Proceedings | bibo:Proceeding | bibo:Volume | bibo:Proceeding | swrc:Proceedings | swrc:Proceedings | swrc:Proceedings |
| **Papers** | swrc:InProceedings | swrc:InProceedings, foaf:Document | bibo:Article | swrc:InProceedings | swrc:Publication | swrc:Publication | swc:Paper |
| **Persons** | foaf:Agent | foaf:Person | foaf:Person | foaf:Person | foaf:Person | foaf:Person | foaf:Person |

**Table 6  Number of entities per concept for each solution (Task 1—2014 and 2015 editions).**

|  | Solution 1.1 | | Solution 1.2 | | Solution 1.3 | | Solution 1.4 |
|---|---|---|---|---|---|---|---|
| Year | 2014 | 2015 | 2014 | 2015 | 2014 | 2015 | 2015 |
| Conferences | 21 | 46 |  | 46 |  | 5 | 47 |
| Workshops | 132 | 252 | 14 | 1,393 | 1,516 | 127 | 198 |
| Proceedings | 126 | 243 | 65 | 1,392 | 124 | 202 | 1,353 |
| Papers | 1,634 | 3,801 | 971 | 2,452 | 1,110 | 720 | 2,470 |
| Persons | 2,854 | 6,700 | 202 | 6,414 | 2,794 | 3,402 | 11,034 |

**Table 7  Statistics about the produced dataset (Task 1—2014 and 2015 editions).**

|  | Solution 1.1 | | Solution 1.2 | | Solution 1.3 | | Solution 1.4 |
|---|---|---|---|---|---|---|---|
| Year | 2014 | 2015 | 2014 | 2015 | 2014 | 2015 | 2015 |
| dataset size | 1.5 M | 25 M | 1.7 M | 7.2 M | 2.7 M | 9.1 M | 9.7 M |
| # triples | 32,088 | 177,752 | 14,178 | 58,858 | 60,130 | 62,231 | 79,444 |
| # entities | 4,770 | 11,428 | 1,258 | 11,803 | 9,691 | 11,656 | 19,090 |
| # properties | 60 | 46 | 43 | 23 | 45 | 48 | 23 |
| # classes | 8 | 30 | 5 | 10 | 10 | 19 | 6 |

in the Semantic Publishing Challenge only in 2015. It combines different page analysis methods, i.e., layout analysis and visual and textual feature classification to analyze the rendered pages, rather than their code. The solution is quite generic but requires domain/case-specific actions in certain phases (model building step).

*Task 2.* There were eight distinct solutions proposed to address Task 2 in the 2015 and 2016 editions of the challenge. Three participated in both editions, three only in 2015 and two only in 2016. As the definition of Task 2 changed fundamentally from 2014 to 2015, the only solution submitted for Task 2 in 2014 (*Bertin & Atanassova, 2014*) is not comparable to the 2015 and 2016 solutions and therefore not discussed here. All solutions for Task 2—except for the one of 2014—are briefly introduced here and summarized in Tables 4, Tables 8–11. Tables 9 and 10 provide details about the methodologies and approach each solution followed. Table 11 summarizes details regarding the implementation and its components each solution employed to address Task 2. Table 4 summarizes the model and vocabularies/ontologies each solution used (both for Task 1 and Task 2), whereas Table 8 provides statistics regarding the dataset schema/entities and triples/size each solution produced (again both for Task 1 and Task 2).

*Solution 2.1.* *Tkaczyk & Bolikowski (2015)* relied on CERMINE (http://cermine.ceon.pl/), an open source system for extracting structured metadata and references from scientific publications published as PDF files. It has a loosely captured architecture and a modular workflow based on supervised and unsupervised machine-learning techniques, which simplifies the systems adaptation to new document layouts and styles. It employs an enhanced Docstrum algorithm for page segmentation to obtain the document's hierarchical structure, Support Vector Machines (SVM) to classify its zones, heuristics and regular

**Table 8** Statistics about the produced dataset (Task 2—2015 and 2016 editions).

| | Sol 2.1 | Sol 2.2 | | Sol 2.3 | Sol 2.4 | Sol 2.5 | Sol 2.6 | Sol 2.7 | Sol 2.8 |
|---|---|---|---|---|---|---|---|---|---|
| Year | 2015 | 2015 | 2016 | 2016 | 2015 | 2015 | 2015 | 2016 | 2016 |
| dataset size | 2.6 M | 1.5 M | 285 | 184K | 3.6 M | 2.4 M | 17 M | 152 | 235 |
| # triples | 21,681 | 10,730 | 2,143 | 1,628 | 15,242 | 12,375 | 98,961 | 1,126 | 1,816 |
| # entities | 4,581 | 1,300 | 334 | 257 | 3,249 | 2,978 | 19,487 | 659 | 829 |
| # properties | 12 | 23 | 23 | 15 | 19 | 21 | 36 | 571 | 23 |

expressions for individual and Conditional Random Fields (CRF) for affiliation parsing and thus to identify organization, address and country in affiliation. Last, K-Means clustering was used for reference extraction to divide references zones into individual reference strings.

*Solution 2.2.* *Klampfl & Kern (2015)* and *Klampfl & Kern (2016)* implemented a processing pipeline that analyzes a PDF document structure incorporating a diverse set of machine learning techniques. To be more precise, they employ unsupervised machine learning techniques (Merge-&-Split algorithm) to extract text blocks and supervised (Max Entropy and Beam search) to extend the document's structure analysis and identify sections and captions. They combine the above with clustering techniques to obtain the article's hierarchical table of content and classify blocks into different meta-data categories. Heuristics are applied to detect the reference section and sequence classification to categorize the tokens of individual references to strings. Last, Named Entity Recognition (NER) is used to extract references to grants, funding agencies, projects, figure and table captions.

*Solution 2.3.* *Nuzzolese, Peroni & Reforgiato Recupero (2015)* and *Nuzzolese, Peroni & Recupero (2016)* relied on the Metadata And Citations Jailbreaker (MACJa IPA) in 2015, which was extended to the Article Content Miner (ACM) in 2016. The tool integrates hybrid techniques based on Natural Language Processing (NLP, Combinatory Categorial Grammar, Discourse Representation Theory, Linguistic Frames), Discourse Reference Extraction and Linking, and Topic Extraction. It also employs heuristics to exploit existing lexical resources and gazetteers to generate representation structures. Moreover, it incorporates FRED (http://wit.istc.cnr.it/stlab-tools/fred), a novel machine reader, and includes modules to query external services to enhance and validate data.

*Solution 2.4.* *Sateli & Witte (2015)* and *Sateli & Witte (2016)*, relying on LODeXporter (http://www.semanticsoftware.info/lodexporter), proposed an iterative rule-based pattern matching approach. The system is composed of two modules: (i) a text mining pipeline based on the GATE framework to extract structural and semantic entities. It leverages existing NER-based text mining tools to extract both structural and semantic elements, employing post-processing heuristics to detect or correct the authors affiliations in a fuzzy manner, and (ii) a LOD exporter, to translate the document annotations into RDF according to custom rules.

Dimou et al. (2017), *PeerJ Comput. Sci.*, DOI 10.7717/peerj-cs.105

**Table 9** Task 2 solutions: their primary analysis methods, their methodologies (i) in general as well as with respect to (ii) extraction, (iii) text recognition and (iv) use of machine learning techniques, and evaluation results.

| | Solution 2.1 | Solution 2.2 | Solution 2.3 | Solution 2.4 | Solution 2.5 | Solution 2.6 | Solution 2.7 | Solution 2.8 |
|---|---|---|---|---|---|---|---|---|
| **Publications** | *Tkaczyk & Bolikowski (2015)* | *Klampfl & Kern (2016)* | *Nuzzolese, Peroni & Recupero (2016)* | *Sateli & Witte (2016)* | *Kovriguina et al. (2015)* | *Ronzano et al. (2015)* | *Ahmad, Afzal & Qadir (2016)* | *Ramesh et al. (2016)* |
| | – | *Klampfl & Kern (2015)* | *Nuzzolese, Peroni & Reforgiato Recupero (2015)* | *Sateli & Witte (2015)* | – | – | – | – |
| **Primary Analysis** | | | | | | | | |
| structure-based | ✓ | ✓ | | | | ✓ | ✓ | ✓ |
| linguistic-based | | ✓ | ✓ | ✓ | ✓ | ✓ | ✓ | |
| presentation-based | ✓ | | | | ✓ | ✓ | | ✓ |
| **Methodology** | | | | | | | | |
| workflow | parallel pipelines | parallel pipelines | single pipeline | iterative approach | single pipeline | single pipeline | single pipeline | layered approach |
| external services | | ✓ | ✓ | ✓ | | ✓ | | |
| **Extraction** | | | | | | | | |
| PDF-to-XML | ✓ | ✓ | | ✓ (2016) | | | ✓ | ✓ |
| PDF-to-HTML | | | | | ✓ | | | |
| PDF-to-text | | | ✓ | ✓ (2015) | | ✓ | ✓ | |
| **Machine Learning** | | | | | | | | |
| supervised | ✓ | ✓ | ✓ | ✓ | | ✓ | | ✓ |
| unsupervised | ✓ | ✓ | | | | | | |
| CRF | ✓ | ✓ | | | | | | ✓ |
| **Text recognition** | | | | | | | | |
| NLP/NER | | ✓ | ✓ | ✓ | ✓ | ✓ | | |
| heuristics | ✓ | ✓ | ✓ | ✓ | ✓ | ✓ | ✓ | ✓ |
| regEx | ✓ | ✓ | ✓ | ✓ | ✓ | ✓ | | ✓ |
| **Evaluation** | | | | | | | | |
| best performing | ✓ (2015) | | | | | | ✓ (2016) | |
| most innovative | | ✓ (2016) | | ✓ (2015) | | | | |

Dimou et al. (2017), *PeerJ Comput. Sci.*, DOI 10.7717/peerj-cs.105

**Table 10** Task 2 solutions: how they address different subtasks to accomplish Task 2.

| Information to extract | Solution 2.1 | Solution 2.2 | Solution 2.3 | Solution 2.4 | Solution 2.5 | Solution 2.6 | Solution 2.7 | Solution 2.8 |
|---|---|---|---|---|---|---|---|---|
| document structure | enhanced docstrum | max entropy, merge & split, clustering | NLP to break the text down in sections & sentences | span between Gazetteer's segment headers | font characteristics, text position | rule-based iterative PDF analysis | heuristics on titles, capital-case and style | level I & II CRF |
| fragments' classification | SVM | supervised ML | Stanford CoreNLP & NLTK | Gazetteer | font-based blocks & sorting | structural features, chunk-& sentence-based SVM | pattern-matching | level II CRF |
| authors | SVM (Lib-SVM) | unsupervised ML & classification | heuristics, NER, CoreNLP | Gazetteer's person first names | e-mail 1st part frequent patterns & string comparison | layout info, ANNIE, external repos | from plain text: start/end identifiers return character | level III CRF |
| affiliations | CRF | unsupervised ML & classification | NER, statistical rules, patterns | organizations names rules patterns | e-mail 2nd part frequent patterns & string comparison | ANNIE, external repos | from plain text: start/end identifiers return character | level III CRF, affiliation markers, POS, NER |
| funding | ✗ | NER, sequence classification | 'Acknowledgments' section, regEx, number or identifier | 'Acknowledgments' section, upper-initial word token or name of organization | 'Acknowledgments' section, string-matching: 'support— fund— sponsor', etc. | manual JAPE grammars | 'Acknowledgments' section, string matching: 'the'…'project', etc. | level II CRF |
| references | CRF | geometrical block segmentation | ParseCit Cross-Ref | hand-crafting rules for multiple cases | Heuristics on 'References' section | external services | n/a | level III CRF (even though n/a in 2016) |
| ontologies | ✗ | n/a | match named entities to indexed ontologies | root tokens of ontology names | 'Abstract' stop-list of acronyms | JAPE grammars | n/a | n/a |
| tables & figures | n/a | max entropy, merge & split | ✗ | 'Table'— 'Figure— Fig' trigger words | n/a | n/a | heuristics on captions, string matching | level II CRF |
| supplementary material | n/a | max entropy, merge & split | ✗ | heuristics on links | n/a | n/a | heuristics on links and string matching | ✗ |

**Notes.**

n/a, stands for subtasks that were not required the year the solution participated in the challenge; ✗, stands for subtasks that were not addressed by a certain solution.

Dimou et al. (2017), *PeerJ Comput. Sci.*, DOI 10.7717/peerj-cs.105

**Table 11  Implementation details for Task 2 solutions.**

| | Solution 2.1 | Solution 2.2 | Solution 2.3 | Solution 2.4 | Solution 2.5 | Solution 2.6 | Solution 2.7 | Solution 2.8 |
|---|---|---|---|---|---|---|---|---|
| **Implementation language** | | | | | | | | |
| C++ | | | | | | | | ✓ |
| Java | ✓ | ✓ | ✓ | ✓ | | ✓ | ✓ | ✓ |
| Python | | ✓ | ✓ | | ✓ | | | |
| **PDF character extraction** | | | | | | | | |
| Apache PDFBox[1] | | ✓ | | | | | ✓ | ✓ |
| iText[2] | ✓ | | | | | | | |
| Poppler[3] | | | | | | ✓ | | |
| PDFMiner[4] | | | ✓ | | ✓ | | | |
| PDFX[5] | | | | ✓ (2016) | | ✓ | ✓ | |
| Xpdf[6] | | | | ✓ (2015) | | | | |
| **Intermediate representation** | | | | | | | | |
| HTML | | | | | ✓ | | | |
| JSON | | | ✓ | | | | | |
| text | | ✓ | | ✓ | ✓ | | ✓ | |
| XML | ✓ (NLM JATS) | | | | | ✓ | ✓ | ✓ (NLM JATS) |
| **External components** | | | | | | | | |
| CrossRef API | | | ✓ | | | ✓ | | |
| DBpedia Spotlight[7] | | | | ✓ | | ✓ | | |
| GATE | | ✓ | | ✓ | | ✓ | | |
| ANNIE[8] | | | | ✓ | | ✓ | | |
| FreeCite | | | ✓ | | | ✓ | | |

Dimou et al. (2017), *PeerJ Comput. Sci.*, DOI 10.7717/peerj-cs.105

**Table 11** (*continued*)

| | Solution 2.1 | Solution 2.2 | Solution 2.3 | Solution 2.4 | Solution 2.5 | Solution 2.6 | Solution 2.7 | Solution 2.8 |
|---|---|---|---|---|---|---|---|---|
| others | GRMM[9], LibSVM[10], Mallet[11] | crfsuite[12], OpenNLP[13], ParsCit[14], | FRED, Stanford CoreNLP[15], NLTK[16], (Word-Net[17], BabelNet[18]) | DBpedia SPARQL end-point | Grab spider[19], Beautiful-Soup[20] | Bibsonomy[21], FundRef[22], | EDITpad Pro[23], | Stanford NERTagger[24], CRF++[25], CoNLL[26], JATS2RDF[27] |
| **(Open Source) License** | AGPL-3.0 | AGPL-3.0 | not specified | LGPL-3.0[28] | MIT | not specified | not specified | not specified |

**Notes.**

[1] Apache PDFBox, https://pdfbox.apache.org/.

[2] iText, http://itextpdf.com/.

[3] Poppler, https://poppler.freedesktop.org/.

[4] PDFMiner, http://www.unixuser.org/~euske/python/pdfminer/.

[5] PDFX, http://cs.unibo.it/save-sd/2016/papers/html/pdfx.cs.man.ac.uk.

[6] Xpdf, http://www.foolabs.com/xpdf/.

[7] DBpedia Spotlight, http://spotlight.dbpedia.org/.

[8] ANNIE, https://gate.ac.uk/sale/tao/splitch6.html.

[9] GRMM, http://mallet.cs.umass.edu/grmm/.

[10] LibSVM, https://www.csie.ntu.edu.tw/~cjlin/libsvm/.

[11] Mallet, http://mallet.cs.umass.edu/

[12] crfsuite, http://www.chokkan.org/software/crfsuite/.

[13] OpenNLP, https://opennlp.apache.org/.

[14] ParsCit, http://wing.comp.nus.edu.sg/parsCit/.

[15] Stanford CoreNLP, http://stanfordnlp.github.io/CoreNLP/.

[16] NLTK, http://www.nltk.org/.

[17] WordNet, https://wordnet.princeton.edu/.

[18] BabelNet, http://babelnet.org/.

[19] Grab spider, http://grablib.org/.

[20] BeautifulSoup, http://www.crummy.com/software/BeautifulSoup/.

[21] Bibsonomy, http://www.bibsonomy.org/help/doc/api.html.

[22] FundRef, http://www.crossref.org/fundingdata/.

[23] EDITpad Pro, https://www.editpadpro.com/.

[24] Stanford NERTagger, http://nlp.stanford.edu/software/CRF-NER.shtml.

[25] CRF++, https://taku910.github.io/crfpp/.

[26] CoNLL, http://www.cnts.ua.ac.be/conll2000/chunking/.

[27] JATS2RDF, https://github.com/Klortho/eutils-org/wiki/JATS2RDF.

[28] LGPL-3.0, https://opensource.org/licenses/lgpl-3.0.html.

*Solution 2.5. Kovriguina et al. (2015)* relies on a rule-based and pattern matching approach, implemented in Python. Some external services are employed for improving the quality of the results (for instance, DBLP for validating authors data), as well as regular expressions, NLP methods and heuristics for HTML document style and standard bibliographic description. It also relies on an external tool to extract the plain text from PDFs.

*Solution 2.6. Ronzano et al. (2015)* extended their framework used for Task 1 (and indicated as Solution 1.3 above) to extract data from PDF as well. Their linear pipeline includes text processing and entity recognition modules. It employs external services for mining PDF articles and heuristics to validate, refine, sanitize and normalize the data. Moreover, linguistic and structural analysis based on chunk-based & sentence-based SVM classifiers are employed, as well as enrichment by linking with external resources such as Bibsonomy, DBpedia Spotlight, DBLP, CrossRef, FundRef & FreeCite.

*Solution 2.7. Ahmad, Afzal & Qadir (2016)* proposed a heuristic-based approach that uses a combination of tag-/rule-based and plain text information extraction techniques combined with generic heuristics and patterns (regular expressions). Their approach identifies patterns and rules from integrated formats.

*Solution 2.8. Ramesh et al. (2016)* proposed a solution based on a sequential three-level Conditional Random Fields (CRF) supervised learning approach. Their approach follows the same feature list as *Klampfl & Kern (2015)*. However, they extract PDF to an XML that conforms to the NLM JATS DTD, and generate RDF using an XSLT transformation tool dedicated for JATS.

### Tasks evaluation

The evaluation of the submitted solutions was conducted in a transparent and objective way by measuring precision and recall. To perform the evaluation, we relied on (i) a gold standard and (ii) an evaluation tool which was developed to automate the procedure.

*Gold standard.* The gold standard used for each task's evaluation was generated *manually*. It consisted of a set of CSV files, each corresponding to the output of one of the queries used for the evaluation. Each file was built after checking the original sources—for instance HTML proceedings in case of Task 1 and PDF papers for Task 2—and looking for the output of the corresponding query; then, it was double-checked by the organizers. Furthermore, we also made available the gold standard to the participants (after their submission) so as they have the chance to report inaccuracies or inconsistencies. The final manually-checked version of the CSV files was used as input for the evaluation tool.

*Evaluation tool.* The evaluation tool (SemPubEvaluator, https://github.com/angelobo/ SemPubEvaluator) compares the queries output provided by the participants (in CSV) against the gold standard and measures precision and recall. It was not made available to the participants after the 2014 edition, it was only made available after the 2015 edition, while it was made available already by the end of the training for the 2016 edition. This not

only increased transparency but also allowed participants to refine their tools and address output imperfections, increasing this way the quality of their results.

# BEST PRACTICES FOR CHALLENGE ORGANIZATION

In this section we discuss lessons learned from our experience in organizing the challenge and from (even unexpected) aspects that emerged while running the challenge. This section presents the lessons learned by looking at the solutions and data produced by the participants. We have grouped the lessons in categories for clarity, even though there is some overlap between them.

Moreover, we validated our lessons learned with respect to other Semantic Web Evaluation Challenges, aiming to assess whether the lessons learned from the Semantic Publishing Challenge are transferable to their settings too. Besides the Semantic Publishing Challenge, another five challenges are organized in the frame of the Semantic Web Evaluation Challenges track at the ESWC Semantic Web Conference (cf. 'State of the art on previously organized challenges' section). To validate our challenge's lessons learned, we conducted a survey, which we circulated among the organizers of the different Semantic Web Evaluation Challenges. One organizer per challenge filled in the questionnaire, providing representative answers for the respective challenge. Based on our survey's results, we distill generic best practices that could be applied to similar events. Our lessons learned are outlined in this section, together with their validation based on the other challenges, as well as the corresponding distilled best practices.

## Lessons learned from defining tasks

For the Semantic Publishing Challenge, it was difficult to define appealing tasks that bridge the gap between building up initial datasets and exploring possibilities for innovative semantic publishing. Therefore, as discussed in 'Semantic Publishing Challenge: 2014–2016', we refined the challenge's tasks over the years according to the participants' and organizers' feedback.

### Task continuity

*Lesson.* In the case of the Semantic Publishing Challenge, the first edition's tasks were well perceived by potential participants and all of them had submissions. In the second edition (2015), in fact, the challenge was re-organized aiming at committing participants to re-submitting overall improved versions of their first edition's submissions. Results were positive, as the majority of the participants of the first edition competed in the second one too. Therefore, task continuity is a key aspect of the Semantic Publishing Challenge, whose tasks in every year are broadly the same as the previous year's edition, allowing participants to reuse their tools to adapt to the new call after some tuning.

*Validation.* Three of the other four Semantic Web Evaluation Challenges have also been organized for several times. Table 1 shows the sustainability of the challenges considering recency and regularity of revisions over their lifetimes. Task continuity was embraced in all challenges by their participants, who not only resubmitted their solutions but also showed

continuously improved performance for all three challenges that had multiple editions, according to the organizers' answers to our survey.

*Best practice.* Tasks should be continued over the course of different editions. Nevertheless, they should be adjusted to pose new challenges that allow the authors of previous editions' submissions to participate again in the challenge, thus offering them incentives to improve their solution, without excluding though new submissions at the same time.

### Distinct tasks

*Lesson.* The initial goal of the Semantic Publishing Challenge was to explore a larger amount of information derived from CEUR-WS.org data and to offer a broad spectrum of alternative options for potential participants but, in retrospect, such heterogeneity proved to become a limitation. One of the main problems we faced was that some of the queries classified under the same task were cumbersome for the participants. For instance, in particular the submissions to Task 2—extraction from XML and PDF—showed an unexpectedly low performance. The main reason, in our opinion, is that the task was actually composed of two sub-tasks that required different tools and technologies: some queries required participants to basically map data from XML/PDF to RDF, while the others required additional processing of the content. Potential participants were discouraged to participate as they only felt competitive for the one and not for the other. A sharper distinction between tasks would have been more appropriate. In particular, it is important to separate tasks on plain data extraction from those on natural language processing and semantic analysis.

*Validation.* According to the results of our survey, the Semantic Web Evaluation Challenges were designed with more than one task, more precisely, on average three tasks per challenge. In addition, all the individual tasks of the challenges were defined related to each other but independently at the same time, so that participants could take part in all or some of the tasks. Nevertheless, only two challenges had submissions for all tasks, while three out of five challenges lacked submissions only for one task. All challenges though, according to our survey, split the tasks considering the required competencies to accomplish them. Three out of five challenges even distinguish the training dataset used by each task to render the different tasks even more distinct. This contributes to enabling participation in certain tasks, while more challenging tasks or tasks of different nature are isolated. Thus, participants are not discouraged from participating if they are not competent for these parts; they can still participate in the tasks where they feel competent.

*Best practice.* Splitting tasks with a clear and sharp distinction of the competencies required to accomplish them is a key success factor. Task should be defined taking into consideration the technology, tools and skills required to accomplish them.

### Participants involvement

*Lesson.* One of the incentives of the challenge's successive editions was to involve participants in the tasks' definition, because potential tasks or obstacles might be identified more easily, if not intuitively, by them. However, even though we collected feedback from

previous years' participants when designing the tasks, we noticed that such a preliminary phase was not given enough attention. Even though participants provided feedback immediately after the challenge was completed they were not equally eager to give feedback when they were asked just before the new edition was launched. Talking to participants, in fact, helped us to identify alternative tasks.

*Validation.* It is common practice that challenge organizers ask for the participants' feedback. According to our survey three out of four challenges (including Semantic Publishing Challenge) which had more than one submission took into consideration the participants' feedback to adjust the tasks or to define new.

*Best practice.* Exploiting participants feedback and involving them in the task definition creating a direct link between different editions is a key success factor. The participants' early feedback can help to identify practical needs and correspondingly shape and adjust tasks. Tasks proposed or emerged from the community can be turned into an incentive to participate.

### Community traction

*Lesson.* Although the challenge was open to everyone from industry and academia, we originally expected participants from the Semantic Web community. However, the submitted solutions include participants with completely different research focus areas, even without any Semantic Web background. This changed our perception of the core communities in the challenge. In future, one might therefore consider defining a cross-domain task, e.g., using a dataset of publications from the biomedical domain.

*Validation.* Evaluating the scientific profiles of participants and the submitted solutions highlights the diversity of professions. The participants of Task 2 are mainly active researchers in the fields of NLP (Natural Language Processing), Text Mining, and Information Retrieval. Submissions to Task 1 are mostly from the Linked Data and semantic publishing communities, addressing various subjects of interest such as User Modeling, Library Science, and Artificial Intelligence. This diversity of professions was acknowledged while inviting the members of the challenge's program committee, and during the process of assigning them as reviewers to submissions.

*Best practice.* Defining independent tasks and using datasets related to other fields of study can build a bridge across disciplines. The use case dataset contains data about computer science publications, and the super-event of the Semantic Publishing Challenge series, the ESWC conference, is highly ranked, and thus of potential interest to a wide audience, but focused on a dedicated sub-field of computer science. This choice of subject potentially restricts the target audience and the publicity of the challenge; however, with a slight shift of any of these, it becomes possible to involve other research communities.

## Lessons learned from building training and evaluation datasets

The training and output dataset definition are also crucial parts when organizing a challenge. In the Semantic Publishing Challenge case, we experimented with (i) maintaining the same

training and output dataset, as well as the same tasks, as in the case of Task 1, and (ii) modifying the dataset but keeping almost the same tasks, as in the case of Task 2 and 3. This way, we bridged the gap between building up initial datasets and exploring possibilities for innovative semantic publishing. As mentioned in 'Semantic Publishing Challenge: 2014–2016' section, we refined both the datasets and their corresponding tasks over the years according to the participants' and organizers' feedback.

### Dataset continuity

*Lesson.* We noticed benefits of not only continuing the same tasks but also using the same datasets across multiple editions of the challenge. In Task 1 of each edition, we evolved training and evaluation datasets based on the same data source over the three years. Participants were able to reuse their existing tools and extend the previously-created knowledge-bases with limited effort. However, for the other tasks, whose datasets were not equally stable, we had to rebuild the competition every year without being able to exploit the past experience. Once solutions were submitted for Task 2 though and it was repeated with the same dataset in 2016 as in 2015, the Semantic Publishing Challenge immediately gained corresponding profit as for Task 1, as the majority of the submitted solutions were resubmitted. This did not happen with Task 3, which did not gain traction in the first place and changing the training dataset and tasks did not attract submissions. Therefore, the "continuity" lesson is equally applicable to tasks as well as to datasets.

*Validation.* Dataset continuity is not as persistent as task continuity for most challenges, but it still occurs. To be more precise, most challenges in principle reuse the same datasets across different editions: two of the four Semantic Web Evaluation Challenges with multiple editions reused the same dataset, while the other two did the same except for one of their editions, where a different dataset was considered, albeit one of the same nature.

*Best practice.* Same datasets should be continuously reused over the course of different editions. Nevertheless, eventually substituting them by another dataset of the same nature, where the same tasks and tools are equally applicable, does not harm the challenge.

### Single dataset for all tasks

*Lesson.* Similarly, we observed that it is valuable to use the same dataset for multiple tasks. For instance, in the Semantic Web Challenge case, completely different datasets were used for Task 1 and 2 for the first edition, but complementary datasets were used for the same tasks during the second and third edition, while Task 3 considered the previous year's output of Task 1.

The participants can extend their existing tools to compete for different tasks, with limited effort. This also opens new perspectives for future collaboration: participants' work could be extended and integrated in a shared effort for producing useful data. It is also worth highlighting the importance of such uniformity for the organizers. It reduces the time needed to prepare and validate data, as well as the risk of errors and imperfections. Last but not least, it enables designing interconnected tasks and producing richer output.

*Validation.* All four Semantic Web Evaluation Challenges with multiple editions used the same dataset or subsets of it for all different tasks of the challenge.

*Best practice.* It is clearly beneficial for the challenge to consider the same dataset for all tasks.

### Exhaustive output dataset description

*Lesson.* An aspect that was underestimated in the first editions of the Semantic Publishing Challenge was the training and output dataset description. While we completely listed all data sources, we did not provide enough information on the expected output: we went into details for the most relevant and critical examples, but we did not provide the exact expected output for all cases in the training dataset. Such information should have been provided, as it directly impacts the quality of the submissions and helps participants to refine their tools.

*Validation.* According to the survey results, the other Semantic Web Evaluation Challenges seem to share the same principle about the exhaustive description of the expected output dataset. To be more precise, only one of the Semantic Web Evaluation Challenges does not provide a detailed and exhaustive description of the expected output.

*Best practice.* Exhaustive and detailed description of both the training and evaluation dataset is required, as it affects the submissions' quality and helps participants to refine their tools.

## Lessons learned from evaluating results

All three editions of the Semantic Publishing Challenge shared the same evaluation procedure (see 'Tasks evaluation' for details). However, it presented some weaknesses, especially in the first two editions, which we subsequently addressed. Three lessons are derived from the issues that are explained below.

### Entire dataset evaluation

*Lesson.* Even though we asked participants to run their tools on the entire evaluation dataset, we considered only a subset for the final evaluation. The subset has been randomly selected from clusters representing different cases, which participants were required to address. On the one hand, since the subset was representative of these cases, we received a fair indication of each tool capabilities. On the other hand, some submissions were penalized as their tool could have worked well on other values, which were not taken into account for the evaluation. In the second edition, we tried to resolve this issue by increasing the number of evaluation queries, without reaching the desired results though, but causing instead some additional overhead to the participants. In the third edition, we reduced the number of evaluation queries, but we radically increased their coverage to assure that the greatest part of the dataset (or even the whole dataset) is covered.

*Validation.* Our lesson learned was validated by our survey in this case too. Only one of the Semantic Web Evaluation Challenges does not take into consideration the entire dataset for the evaluation.

*Best practice.* The evaluation method should cover the entire evaluation dataset to be fair, to avoid bias and to reinforce submissions to maintain a high quality across the entire dataset.

### Disjoint training and evaluation dataset

*Lesson.* During the first two editions of the Semantic Publishing Challenge, the evaluation dataset was a superset of the training one. This may have resulted in some over-training of the tools, and caused imbalance in the evaluation, as certain tools performed very well for the training dataset but not for the entire dataset. In an effort to avoid this, we made the training and evaluation datasets disjoint for the third edition of the Semantic Publishing Challenge. It is more appropriate to use completely disjoint datasets, as a solution to avoid over-trained tools.

*Validation.* Our lesson learned regarding disjoint training and evaluation datasets was validated by the other challenge organizers. Only one of the Semantic Web Evaluation Challenges considers an evaluation dataset which is a subset of the training dataset. All the others consider disjoint training and evaluation datasets.

*Best practice.* The training and evaluation dataset should be disjoint to avoid over-trained tools.

### Available evaluation tool

*Lesson.* The evaluation was totally transparent and all participants received detailed feedback about their scores, together with links to the open source tool used for the final evaluation. However we were able to release the evaluation tool only after the challenge for the last two editions. The evaluation tool was not made available after the 2014 edition, it was only made available after the 2015 edition, while it was made available by the end of the training for the 2016 edition. It is instead more meaningful to make it available during the training phase, as we did for the challenge's third edition. Participants can then refine their tool and improve the overall quality of their output. Moreover, such an approach reduces the (negative) impact of output imperfections. Though the content under evaluation was normalized and minor differences were not considered as errors, some imperfections were not expected and were not handled in advance. Some participants, for instance, produced CSV files with columns in a different order or with minor differences in the IRI structure. These all could have been avoided if participants had received feedback during the training phase, with the evaluation tool available as a downloadable stand-alone application or as a service.

*Validation.* Our lesson learned regarding the availability of the evaluation tool was also validated by our survey. To be more precise, all the Semantic Web Evaluation Challenges

make the evaluation tool available to the challenge participants. There is only one that does not, but only because there is no evaluation tool.

*Best practice.* The evaluation tool should be made available to the participants as early as possible while the participants are still working with the training dataset and fine tuning their approaches.

## Lessons learned from expected output use and synergies

In all three editions of the Semantic Publishing Challenge, the potential use of the expected output was clearly stated in the call, but not the output dataset license; it was up to the participants to choose one. Moreover, the challenge was disseminated and supported thanks to synergies with other events. In this section, we outline lessons learned regarding how the expected use of the challenge output and synergies reflect on the challenge perspective, also on the participants and their submissions.

### Expected output use

*Lesson.* The uppermost goal of the Semantic Publishing Challenge was to obtain the best output dataset. To achieve that, it is required to identify the best performing tool, namely the tool that actually produces the best output dataset. This tool—or a refined version—is subsequently used to generate the RDF representation of the whole CEUR-WS.org corpus[2]. The fact that the submitted tools are expected to be reused becomes a critical issue: participants' submission should not only target the challenge, but they should produce an output that is directly reusable. Therefore, it is in fact critical to state how the results of the challenge will be eventually used, in order to encourage and motivate participants.

*Validation.* Three out of the other four Semantic Web Evaluation Challenges do clearly mention the expected output use, as the Semantic Publishing Challenge does too.

*Best practice.* The expected output use and conditions should be explicitly specified in advance.

### License

*Lesson.* The incentive to organize the Semantic Publishing Challenge was to reuse the output dataset. Thus, having the permission to do so, which is specified through the dataset license, but also to reuse the tool that produces this output to systematically generate the CEUR-WS.org dataset, is of crucial importance. Particular attention should be given to the licensing of the output produced by the participants. We did not explicitly say which license the submitted solutions should have: we just requested from participants to use an open license on data (at least as permissive as the source of data) and we encouraged open-source licenses on the tools (but not mandatory). Most of the participants did not declare which exact license applies to their data. This is an obstacle for its reusability: especially when data come from heterogeneous sources (e.g., paper full texts copyrighted by the individual authors, as well as metadata copyrighted by the workshops' chairs) and are heterogeneous in content and format, as in the case of CEUR-WS.org, it is very important to provide an explicit representation of the licensing information.

[2]The extraction tool's integration in the CEUR-WS.org production workflow is still in progress but expected to conclude in 2016.

*Validation.* Like the Semantic Publishing Challenge, none of the other Semantic Web Evaluation Challenges specified the tool or output dataset license. As a result, none of the submitted solutions provided any licensing information, apart from one challenge where some of the submitted solutions provided licensing information. Even though all Semantic Web Evaluation Challenges follow the same practice of not specifying the output dataset potential license, it becomes obvious based on the results that explicitly specifying it is important if the challenge output is desired to be reused.

*Best practice.* The output dataset license should be explicitly requested to be provided for each one of the submitted solutions. Moreover, participants should be advised to respectively specify their tools' licensing information, to enable inference of their potential reusability.

### Conflicts and synergies

*Lesson.* Based on our experience from organizing three editions of the Semantic Publishing Challenge, we realized that the dissemination should happen in a targeted way. To this extent, other events thematically relevant to the challenge are considered important synergies that contribute to generating interest and identifying potential participants: For instance, in the Semantic Publishing Challenge case the fact that the SePublica 2014 workshop on semantic publishing was organized at ESWC 2014 reflected positively on our challenge, since we had fruitful discussions with its participants. Moreover, the fact that results from the first two editions of the Semantic Publishing Challenge (*Vahdati et al., 2016*) were presented at the SAVE-SD workshop on semantics, analytics, visualization and enhancement of scholarly data (SAVE-SD2016 Workshop, http://cs.unibo.it/save-sd/2016/), which was co-located with WWW 2016, contributed to the challenge dissemination's and in particular to an audience both thematically and technologically relevant to the challenge. To the contrary, in 2015, we introduced a task on interlinking and realized possible conflicts with other challenges, like OAEI (Ontology Alignment Evaluation Initiative), which may have resulted in the lack of participation to Task 3—even though Task 3 did not intend to cover the specialized scope of OAEI, but rather put the interlinking task into the scope of a certain use case that merely served in aligning the tasks' outputs among each other and with other datasets in the LOD Cloud. Therefore, we concluded that it is important not only to generate interest but also to identify and avoid potential conflicts.

*Validation.* All Semantic Web Evaluation Challenges collaborate with the ESWC conference, as they are co-located with this event. Besides the main conference, which drives the challenges, it appears that most of them, and in particular the most long-standing ones, also collaborate with other events and, in particular, with other workshops. For instance, the QALD challenge collaborates with the CLEF QA track (http://nlp.uned.es/clef-qa/), and the challenge on Semantic Sentiment Analysis collaborates with the workshop on Semantic Sentiment Analysis (http://www.maurodragoni.com/research/opinionmining/events/), which is also co-organized with ESWC. Last, the OKE challenge collaborates with the Linked Data for Information Extraction workshop (LD4IE) (LD4IE2016 Workshop,

http://web.informatik.uni-mannheim.de/ld4ie2016/LD4IE2016/Overview.html) which, in turn, is co-located with ISWC. According to our survey, none of the other challenges experienced conflicts with further challenges.

*Best practice.* Establish synergies with other events that are thematically and/or technologically relevant to reinforce dissemination and to identify potential participants.

## CHALLENGE SOLUTIONS ANALYSIS

In this section, we discuss observations from the participants' solutions and derive corresponding conclusions that can be used in the Linked Data publishing domain. We group the lessons into four categories: tools, ontologies, data and evaluation process, even though there is some overlap between these aspects.

### Lessons learned from the tools

Valuable indications can be derived by looking at the tools implemented by the participants. In particular, we focus on the software used to address Tasks 1 and 2.

#### *Primary analysis*

*Observation.* The Semantic Publishing Challenge tasks could be addressed by both generic and ad-hoc solutions, as well as different methodologies and approaches; nevertheless, solutions tend to converge.

For Task 1, two out of four solutions primarily consisted of a tool developed specifically for this task, whereas the other two solutions only required task-specific templates or rules to be used within their otherwise generic implementations. In the latter case, Solution 1.2 abstracts the extraction rules from the implementation, whereas Solution 1.4 keeps them inline with the implementation. Those two solutions are generic enough to be adapted even to other domains. Even though solutions were methodologically different, four approaches for dealing with the HTML pages prevailed: (i) *structure-based* (relying on the HTML code/structure), (ii) *layout-based* (relying on the Web page layout), (iii) *linguistic-based*, and (iv) *presentation-based*. **Most tools relied on structured-/layout-based approach** (three out of four) and only one on a partially linguistic-based approach (Solution 1.3).

As far as Task 2 is concerned, there were different methodologies and approaches combined in different ways. The overall picture is summarized in Tables 9 and 10. The nature of the task influenced the proposed solutions. In fact the task was composed of two subtasks: (i) identifying the structural components of the PDF papers and (ii) processing the extracted text. Thus, **some solutions mainly focused on *structure-based* analysis** (five out of eight); others gave more relevance to the *linguistic-based* analysis (three out of eight) for their primary analysis. Last, up to four used the ***linguistic-based* analysis to complement their primary approach**, while two solutions also used formatting styles/rules to increase the quality of their output (*style-based* analysis).

We also observed that most solutions implemented a **modular pipeline**. In particular, the solutions that followed a structure-based analysis had a workflow with a single pipeline, whereas **linguistic-based approaches required parallel or iterative pipelines to address**

**different aspects of the solution and to increase performance**. It is also worth mentioning that two solutions over eight, one being the 2015 most innovative solution, adopted an iterative approach. One of them iterates over the same analysis multiple times to refine the results (Solution 2.4); the other one (Solution 2.8) adopted a layered approach, in which each iteration adds new information to the previously-produced output.

*Conclusion.* The solutions were methodologically different among each other, and modular and hybrid solutions prevailed compared to case-specific ones. This is important as case-specific solutions do not extend beyond the scope of challenges, but generic ones do. It is interesting to note that both 2015 and 2016 the best solutions for Task 2 relied primarily on structure analysis, whereas the most innovative solutions focused on linguistic analysis. This might indicate that further research on linguistic approaches might bring interesting results for optimizing the output of such tasks. A deep analysis of the structure, in fact, made participants capture more information; on the other hand, these approaches were quite straightforward and less innovative. It is interesting, though, to note here that the best performing tool of 2016 grounded its structured-based approach on a prior linguistic analysis, whereas most solutions grounded their linguistic analysis on a prior structure analysis. Thus, hybrid solutions are obviously required but their execution order should not be taken for granted. It is also worth discussing the recall score of the linguistic-based tools: these tools most probably suffer from noisy text extraction. In fact the three solutions (Solution 2.2, Solution 2.3 and Solution 2.4) that mainly rely on linguistic analysis achieved the lowest recall scores both in 2015 and 2016 editions, even though they showed significant improvement in the latter edition.

Similarly, the tool that relied on a linguistic analysis for Task 1 showed significantly lower precision and recall, compared to the other tools, indicating that linguistic-based solutions are not enough, if not supported by a precise structure analysis. Even though the linguistic-based approach was considered a rather innovative way of dealing with Task 1, the evaluation showed that a linguistic-based analysis might not be able to perform as well as a structure-based one.

### Methodologies: extraction, intermediate format and machine learning
*Observation.* Diverse methodologies were employed by the participants to extract and analyze content. There were no prevalent approaches, but some tendencies were observed.

For Task 1, three out of four solutions considered **rules to extract data from the HTML pages**; two of them considered CSS to define the rules, while the other one, which relied on linguistic-based analysis, considered JAPE; the latter solution was based on crawling. Last, **all solutions used regular expressions** at some point of their workflow.

For Task 2, half of the solutions in 2015 but only two out of five in 2016 extracted the text from PDF documents and turned it into plain text. On the contrary, **the majority extracted the text from the PDF files but turned it into XML** (two out of six solutions in 2015 and four out of five in 2016). There was only one solution that used HTML as intermediate format. We noted that, **both in 2015 and 2016, the best performing solutions relied on**

a PDF-to-XML extraction. Moreover, one solution changed from PDF-to-text to PDF-to-XML and indeed performed better in 2016, but we cannot state with high certainty if this was the determining factor. Besides extraction, as far as text analysis is concerned, five solutions in 2015 and four in 2016 relied on supervised Machine Learning. Only two solutions in 2015 and one in 2016 (the same as in 2015) additionally relied on unsupervised Machine Learning to address Task 2. Last, **all solutions employed heuristics and regular expressions**. Five out of six solutions in 2015 employed Natural Language Processing (NLP) and Named Entity Recognition (NER), and those that also participated in 2015 kept NLP/NER in their workflows in 2016.

*Conclusion.* Solutions based on supervised Machine Learning were awarded as the most innovative both in 2015 and in 2016. Therefore, it seems that there is potential on experimenting with supervised Machine Learning approaches to address such a task. Nevertheless, even though the best performing solution in 2015 did use supervised Machine Learning, it is not the case for 2016, which makes us conclude that fundamentally alternative solutions might show good results too. Overall, there is potential for improvement and plenty alternative methodologies can be investigated. The intermediate format used by each solution, on the other hand, had no relevant impact on the final results.

### Source tools

*Observation.* The Semantic Publishing Challenge call did not prescribe (i) the implementation language, (ii) the license, as well as whether the tools should (iii) reuse existing components or external services, and (iv) be open-sourced or not. The participants were allowed to follow their preferred approaches.

Three out of four Task 1 solutions, as shown in Table 3, and seven out of eight Task 2 solutions, as shown in Table 11, **primarily relied on Java-based implementations**. In both cases, the remaining solution relied on Python. Two out of eight solutions for Task 2 complemented their Java-based implementations with Python-based parts. Moreover, as it is observed in Table 3, for Task 1, three out of four solutions **relied on tools totally open-sourced**, while the fourth one, the one that addressed both Task 1 and Task 2, **relied on a stack of tools which are open-sourced**, but the workflow used was not. This is also observed in most tools for Task 2, as shown in Table 11 (six out of the eight solutions).

MIT (http://opensource.org/licenses/mit-license.html) **was the most popular license**, with half solutions for Task 1 using it and one out of eight solutions for Task 2, followed by AGPL-3.0 (https://www.gnu.org/licenses/agpl-3.0.en.html), with two out of eight solutions for Task 2 using it. Last, **half of the solutions incorporated external services** to accomplish the tasks (two out of four for Task 1 and four out of eight for Task 2). The one of the two solutions for Task 1 that used external services was the one that participated both in Task 1 and Task 2. GATE, DBpedia, CrossRef API (http://api.crossref.org/), and FreeCite (http://freecite.library.brown.edu/) are the most used external services.

*Conclusion.* Open-sourced tools prevailed over closed-sourced ones. None of the participants used a totally closed or proprietary software. Most of the them used an

open license, and Java and Python based implementations prevailed both for Task 1 and Task 2. The integration of external services was also a valuable solution for the participants.

## Lessons learned from models and ontologies

In this section, we discuss the different solutions with respect to the data model, the vocabularies and the way they used them to annotate the data.

### Data model

*Observation.* All **Task 1 solutions tend to converge regarding the data model, identifying the same core concepts**: *Conference*, *Workshop*, *Proceedings*, *Papers*, and *Person*. A few solutions covered more details, for instance, Solution 1.1 identified also the concepts of *Invited Papers* and *Proceedings Chair*, while Solution 1.3 captured different types of sessions by identifying additionally the concepts of *Session*, *Keynote Session*, *Invited Session* and *Poster Session*, as well as the concepts of *Organization* and *Topic*. In particular for Task 1, Solution 1.4 domain modeling was inspired by the model used in Solution 1.1, with some simplifications, a practice commonly observed in real Linked Data set modeling.

In contrast, **Task 2 solutions used more heterogeneous data models**. There are six high-level properties identified by all solutions: *identifier*, *type*, *title*, *authors*, *affiliation* and *country*. Other entities were instead described in different ways and with different granularity. That happened, for instance, to the entities *organization*, *funding agency* and *grant*. In certain cases they are identified as separate entities and in other cases their details constitute part of other entities descriptions (and are expressed as data or object properties). The coverage of the data models was also heterogeneous: for the 2016 edition, for instance, not all solutions identify the *sections* and capture the notion of caption of *figures* and *tables*.

*Conclusion.* Based on the aforementioned, we observe a trend of converging in respect to the model the CEUR-WS.org dataset should have according to the submitted solutions. Most solutions converge on the main identified concepts in the data (*Conference*, *Workshop*, *Proceedings*, *Paper* and *Person*) and on the CEUR-WS.org dataset's graph at least for Task 1, namely the publications' metadata. The way the tasks and their corresponding queries are described contributes towards this direction.

### Vocabularies

*Observation.* There is a wide range of vocabularies and ontologies that can be used to annotate scholarly data. Most of the solutions preferred to **(re)use almost the same existing ontologies and vocabularies**, as summarized in Table 4. Six out of twelve solutions for both Task 1 and 2 used the Semantic Web for Research Communities (*swrc*) vocabulary (SWRC, http://swrc.ontoware.org/ontology#), five used the Bibliographic Ontology (*bibo*) vocabulary (bibo, http://purl.org/ontology/bibo/) and three used the Semantic Web Conference (*swc*) vocabulary (SWC, http://data.semanticweb.org/ns/swc/ontology#). Moreover, six solutions used one or more vocabularies of the Semantic Publishing and Referencing Ontologies (*SPAR*, http://www.sparontologies.net/). In particular, five solutions used the FRBR-aligned Bibliographic Ontology (*FaBiO*, http://purl.org/spar/fabio/) ontology, three the Publishing Roles Ontology (*PRO*, http://purl.org/spar/pro/),

three the Document Components Ontology (*DoCO*, http://purl.org/spar/doco/), two the Bibliographic Reference Ontology (*BiRO*, http://purl.org/spar/biro/), two the Funding, Research Administration and Projects Ontology (*FRAPO*, http://purl.org/cerif/frapo/) and one the Functional Requirements for Bibliographic Records (*FRBR*, http://purl.org/spar/frbr/). Besides the domain-specific vocabularies and ontologies, eight solutions used the Dublin Core vocabulary (*dc*, http://purl.org/dc/elements/1.1/ and *dcterms*, http://purl.org/dc/terms/), eight the Friend of a Friend vocabulary (*foaf*, http://xmlns.com/foaf/0.1/), five solutions used the DBpedia ontology (*dbo*, http://dbpedia.org/ontology/), three the VCard (*vcard*, http://www.w3.org/2006/vcard/ns#) and two the *event* (event ontology, http://purl.org/NET/c4dm/event.owl#) and *timeline* (timeline ontology, http://purl.org/NET/c4dm/timeline.owl#) ontologies and *schema.org* (http://schema.org). Last, there were four solutions that **used their own custom vocabularies, in combination with existing ones** in most cases, but only one used barely its custom vocabulary.

In contrast to **Task 1 solutions, which all converged on using same vocabularies and ontologies** intuitively, **Task 2 solutions reused a wider range and relatively different vocabularies and ontologies** to annotate same entities appearing in the same data, which is extracted from PDF documents. This is a consequence of the rather diverse data models considered by different solutions. Interestingly, most Task 2 solutions use sub-ontologies of the *SPAR* ontologies family. Last, most solutions reuse the three most popular vocabularies in the education field according to *Schmachtenberg, Bizer & Paulheim (2014)*. The general purpose vocabularies—such as FOAF—used by the participants are also listed high in the same ranking.

*Conclusion.* It is evident that the spirit of vocabulary reuse gains traction. However, it is interesting that different solutions used the same ontologies to annotate the same data differently (see also 'Annotations' section).

### Annotations
*Observation.* Even though **all solutions used almost the same vocabularies, not all of them used the same vocabulary terms to annotate the same** *entities*. As far as Task 1 is concerned, all solutions only converged on annotating *Persons* using the `foaf:Person` class. For the other main concepts the situation was heterogeneous, as reported in Table 6. A few of them also explicitly annotated *Persons* using the `foaf:Agent` class, even though `foaf:Person` is a subclass of `foaf:Agent`. `foaf:Agent` was also used by one of the solutions during the first edition, but it was then replaced by the more explicit `foaf:Person`. The *Conference* concept was well-captured by all solutions.

It is interesting to note that, **for the first edition, most solutions used relatively generic vocabulary terms**, e.g., `swrc:Event`, `swc:Event` or `swc:OrganizedEvent` to annotate the data. However, **in the second edition, most solutions preferred to use more explicit vocabulary terms for the same concept**, e.g., `swrc:Conference` and `bibo:Conference`, while they also maintained the more generic vocabulary terms for events. The same occurred with the *Paper* concept. The 2014 edition datasets were annotated using more generic vocabulary terms, e.g., `swrc:Publication` or even `foaf:Document`, whereas in

2015 more explicit terms were preferred, such as `swrc:InProceedings` or `bibo:Article`. In particular `swrc:InProceedings` was adopted by three out of four solutions.

In contrast to **Task 1 solutions, which focus on identifying and describing concrete *entities*, Task 2 solutions mainly focus on capturing their *properties*.** This is also evident from the fact that Task 2 solutions rarely provide the entities' types, whereas Task 1 solutions always do, even though this information could be inferred from the properties used. Moreover, Task 2 solutions generate much fewer entities than Task 1 solutions. **All Task 2 solutions use approximately the same number of properties**. It is interesting though to note that solutions that follow in principle the linguistic approach tend to use more predicates, which are more explicit and more descriptive too.

**All solutions have approximately the same number of predicates, but their precision is still not accurate**. Only one of Task 2 solutions (Solution 2.7) has a significantly higher number of predicates compared to the other solutions. This occurs because different URIs are used for the same relationships appearing in different files to annotate the data. For instance, the *section-title* property appears with 37 different URIs, such as the following: [http://ceur-ws.org/Vol-1558/paper5#section-title](http://ceur-ws.org/Vol-1558/paper5#section-title), or [http://ceur-ws.org/Vol-1303/paper_4#section-title](http://ceur-ws.org/Vol-1303/paper_4#section-title). However, such a choice prevents easily identifying same relationships.

*DCMI* **is the vocabulary most frequently used by all solutions for annotating the *identifier* and the *title***. *RDF(S)* is also used for the *title* (represented as `rdfs:label`), as well as for the entities' *types*. For the remaining properties, a wide range of different vocabularies are considered, but they do not converge on their choices. Indicatively: one of the solutions considers `schema:mentions` to describe a citation, whereas other solutions consider `bibo:cites` or `biro:references`. In the same context, some solutions associate authors to papers with the `dcterms:creator` property, whereas others consider `foaf:maker`. Moreover, some solutions indicate the affiliation using the `swrc:affiliation` property, whereas others use `pro:relatesToOrganization`, or some solutions represent the publication year using `swrc:year`, whereas others use `fabio:hasPublicationYear`. Last, it is interesting to note that solutions may even use vocabulary terms that do not exist, such as `swrc:Section`.

*Conclusion.* On the one hand, the more familiar the data publishers get with the data, the more explicit they become with the annotations they use and the more they converge on the choices they make. On the other hand, the way different solutions extract particular properties reflects on the final data model.

## Lessons learned from submitted RDF datasets

In this section, we discuss the different solutions with respect to the RDF dataset they produce.

### Successive submissions improvements

*Observation.* From the first edition to the second edition of the Semantic Publishing Challenge, we noticed that the **participants who re-submitted their solutions had improved the overall dataset**, not only the parts useful to answer the queries. For

instance, all three solutions of Task 1 that had participated in both the 2014 and the 2015 editions modified the way they represented their data, and this resulted in corresponding improvements to the overall dataset.

Indicatively, as far as Task 1 is concerned, Solution 1.2 addressed a number of shortcomings the previous tool's version had, in particular regarding data transformations, which might have influenced their precision improvement. *Heyvaert et al. (2015)* also assessed their mappings' quality to verify the schema is valid with respect to the used vocabularies and ontologies. To address the same issue and avoid inconsistencies in their dataset, Solution 1.1 preferred to align different ontologies' classes and properties, e.g., aligning BIBO to the SWRC ontologies, as SWC already has some dependencies on SWRC.

As far as Task 2 is concerned, some parts of Solution 2.2, for instance, were changed for participating in the 2016 edition. The authors employed different processing steps of their tool, which were not used in the previous edition, e.g., processing section headings, hierarchy and captions, but they also introduced novel aspects driven by the challenge tasks and queries, e.g., extracting links from supplementary material. Among the changes of Solution 2.4, it was the PDF extraction tool used, whose change might have partially contributed to their recall improvement, while a number of additional or new conditional heuristics most probably led to their precision improvement. Overall, it was observed that improvements to extraction might reflect on the solutions' recall, whereas improvements to text analysis on their precision.

*Conclusion.* The improvement of the dataset was evident on some aspects and indeed the results were satisfying, but we still see room for improvement. It is interesting though to note that solutions did not remain focused on improving just the *data extraction* parts of the challenge, but also the *data modeling*, even though the latter is not directly assessed by the challenge.

### Dataset structure

*Observation.* **The different solutions differ significantly with respect to the size of the produced dataset**. This happens for different reasons. Solution 1.1 shows an extraordinary number of triples compared to other solutions. This occurs to a certain extent because each concept is annotated with at least two classes, making one fourth of the dataset to be type declarations. Moreover, they include even annotations that indicate the type of the resource or property on a very low level, namely they use `rdfs:Class`, `rdfs:Property`, as well as `owl:ObjectProperty` or `owl:AnnotationProperty` etc., which counts for almost 2,000 triples of the total dataset. Solution 1.4 also shows a high number of triples. This occurs because the same dataset contains triples describing the structure of the HTML page, as well as triples describing the actual content of the pages. Nevertheless, the main reason that causes the flow of triples is the fact that a new URI is generated each time a concept appears in one of the CEUR-WS.org volumes. For instance, the person *Ruben Verborgh* appears to have 9 URIs, e.g., http://ceur-ws.org/Vol-1034/#RubeniVerborgh for the Vol-1034 proceedings or http://ceur-ws.org/Vol-1184/#RubeniVerborgh for the Vol-1184 proceedings. The person *Christoph Lange* appears to have 15 distinct URIs, e.g., for

[3]The definition of Task 1 was not explicit with regard to whether different persons with the same name (within or across different workshops proceedings volumes) should be assumed to be the same person or not. Our current work towards the release of a consolidated CEUR-WS.org dataset shows that the far majority of same names refers to the same person, which is plausible as CEUR-WS.org focuses on the relatively small computer science community. However, a general solution would be wrong to simply assume that same names mean same persons, whereas a full disambiguation of names would require a lot of information to be taken into account beyond the proceedings' tables of content: the title pages of the PDF papers plus possibly external resources.

[4]Our instructions did not prescribe whether or not participants should assume persons with the same name to be the same. In the reality of the CEUR-WS.org data, there are very few cases in which the same name refers to two different persons, as the data covers the relatively small domain of computer science researchers.

Vol-360 proceedings, the http://ceur-ws.org/Vol-360/#ChristophiLange, or for Vol-1184 proceedings, the http://ceur-ws.org/Vol-1184/#ChristophiLange[3] . Solutions 1.2 and 1.3 are approximately at the same number of triples both for the 2014 and the 2015 editions.

*Conclusion.* There is a very high heterogeneity in the produced datasets; although solutions tend to agree on used vocabularies, their design choices are very different and, as a consequence, the number and organization of the triples is very heterogeneous.

### Coverage

*Observation.* We further noticed that **solutions rarely agree upon the extracted information**. For instance, some skip the extraction of wrong data or certain other information. Overall, we observed significant differences with respect to the number of identified entities per category. The results for Task 1 are summarized in Tables 7 and 6, while the results for Task 2 are summarized in Table 8.

**Produced datasets were very heterogeneous in term of size, number of triples and entities**. As far as Task 1 is concerned, apparently, Solution 1.1 and Solution 1.3 used the individual pages to identify the proceedings, whereas Solution 1.2 and Solution 1.4 used the index page to identify the proceedings, this is the reason that there is so big difference in the number of *Proceedings* entities. The number of identified papers is also significantly different among the different solutions, but in the *Persons* case we observe the greatest variation in terms of numbers because of **different practices of assigning URIs**; a few solutions reuse URIs across different proceedings volumes, others do not[4] .

As far as Task 2 is concerned, **solutions tend to omit certain subtasks and to optimize their performance on others due to the nature of the task**—queries were quite heterogeneous, with a clear distinction, for instance, between the analysis of the structural components and of the textual content of the papers. For instance, in 2015, the best performing solution focused on precisely addressing the subtasks which were related to the document structure and totally omitted queries related to funding and ontologies, as shown in Table 10. Similarly, in 2016, certain solutions completely omitted the queries that were related to supplementary material or tables and pictures captions. Consequently, the dataset size, as well as the number of triples and entities significantly diverge among the solutions.

*Conclusion.* The datasets' heterogeneity is also evident in the amount and type of information each dataset provides. However, the more the solutions improve, the more the solutions converge at least regarding the number of retrieved and/or distinctly identified entities.

## Lessons learned from the solutions with respect to the evaluation

In this section, we discuss the different solutions with respect to the dataset evaluation.

### Ranking

*Observation.* For Task 1, in 2015 the performance ranking of the three tools evolved from 2014 has not changed but their performance has improved except for Solution 1.1, which improved precision but recall remain the same. Disregarding the two queries that were new in 2015, Solution 1.1, which had won the best performance award in 2014, performs almost as well as Solution 1.4.

The trend was slightly different for Task 2: all tools participating in the Challenge for the second time increased their performance, but the overall ranking changed. Solution 2.4 obtained a higher score than Solution 2.2 in 2016, contrarily to what happened in 2015. The position of Solution 2.3 was stable.

*Conclusion.* Continuity helps participants to improve their tools; the overall ranking keeps stable if the tasks (and queries) are kept stable; adjustments to the tasks (and queries) may impact the ranking, favoring one team more than another.

### New and legacy solutions

*Observation.* Task 1 participants both in 2014 and 2015 had an improved version of different aspects of their solution, which resulted in correspondingly improved versions of the final dataset. The new Solution 1.4, which introduced a fundamentally new approach, achieved equally good results as the best solution of 2014. The same trend was evident in Task 2, with a general improvement of all solutions that were re-proposed for the second year (2015 and 2016).

*Conclusion.* Legacy solutions might be able to improve and bring stable and good results, however there is still room for improvement and mainly for fundamentally new ideas that surpass problems that legacy solutions cannot deal with.

### Equal chances

*Observation.* Solution 1.1, the winners of Task 1 in 2014, participated in 2015 with an improved version but did not win. The 2015 winner was a new tool with a brand new approach (Solution 1.4). The same happened for Task 2: in 2016, one winner (Solution 2.7) was a brand-new solution, the other one (Solution 2.2) was an extension and improvement of a legacy solution but did not win the year before.

*Conclusion.* The winners were not the same in subsequent versions of the challenge: creativity won.

## DISCUSSION: CHALLENGE IMPACT ON LINKED DATA QUALITY

In the 'Introduction' section we motivated the Semantic Publishing Challenge as a means of producing high-quality Linked Data. In this section, we assess the potential impact of the challenge on the quality of the Linked Data produced. To be more precise, the quality of the Linked Data produced by the tools submitted has been assessed by comparing the output of a number of prescribed queries against our gold standard and measuring precision and recall, as explained in 'Tasks evaluation' section. Assessing the quality of Linked Data by running queries over it is a common approach, as the comparison of tools by *Zaveri et al. (2016)* confirms, whose recent survey we refer to for a comprehensive review of the state of the art regarding Linked Data quality assessment. Therefore, a challenge designed as the Semantic Publishing Challenge could act as a means to assess the Linked Data quality, and, the better the results, the higher the Linked Data quality is expected to be.

The specific quality metrics that our evaluation setup assesses can be connected to the general quality dimensions (accessibility, intrinsic, contextual and representational) and certain of their corresponding metrics, as they are identified by *Zaveri et al. (2016)*. Moreover, few other quality dimensions' metrics that are not covered by the challenge's evaluation are assessed in the frame of this review. Note that some metrics are applicable for all tasks, whereas others are only for a certain task.

## Accessibility dimensions

The accessibility dimensions involve aspects related to the Linked Data access, authenticity and retrieval (*Zaveri et al., 2016*). Our challenge required participants to make their data available, forcing this way the solutions to cover the availability dimension. Making the data available as an RDF dump was the minimum requirement set by the challenge, covering this way the accessibility of the RDF dumps metric. Participants were also encouraged to publish their data via other Triple Pattern Fragment (TPF) interfaces, such as SPARQL endpoints, but assessing its availability was not part of the challenge's evaluation. Moreover, participants were encouraged to publish their data using a certain license, without being a requirement though, boosting this way the licensing dimension (the corresponding detailed discussion is available in 'License'). While the aforementioned referred to all challenge's tasks, the interlinking dimension was only promoted by Task 3, which, after all, is its actual goal. Overall, even though the submitted solutions only made their datasets available as RDF dumps and did not specify the license, the challenge achieved to enable solutions to achieve the minimum requirement of making the produced datasets accessible. It is evident that, if the challenge had turned high values w.r.t. each of the aforementioned metrics mandatory, the produced dataset accessibility would have been increased.

## Intrinsic dimensions

According to *Zaveri et al. (2016)*, the intrinsic dimensions focus on whether the information correctly, compactly and completely represents the real world and is logically consistent in itself. As the Semantic Publishing Challenge requires SPARQL queries to be executed against the Linked Data produced by the different solutions, the syntactic validity of the dataset is a prerequisite, boosting this way the metrics for syntax error free documents and the absence of malformed datatypes. While our challenge evaluation covers well the syntactic validity, the semantic accuracy is not evaluated. Nevertheless, the metric which is related to the misuse of properties is discussed and assessed in a qualitative way in the 'Annotations' section of this paper, but it is not assessed quantitatively. Similarly, the population completeness, i.e., the percentage of real-world objects of a particular type that are represented in a dataset, is indirectly evaluated on the side. Namely, it is not thoroughly assessed if all real-world entities appear, but to successfully answer the evaluation queries, the population completeness is prerequisite. Moreover, a comparative evaluation of the population completeness is performed in this work (see more detailed discussion at the 'Coverage' section and Tables 7, 8). Last, even though the solutions' dataset consistency dimension could have been evaluated and shed more light to their quality, it was not done by any of the challenge's series so far. All in all, as the challenge was not focused on

assessing the dataset quality, certain metrics of the intrinsic dimension were not covered intentionally, others were indirectly assessed, while a few others were only discussed in this paper. Nevertheless, if it had been intended, the challenge could have covered even more metrics of the intrinsic dimension and could have reinforced the datasets quality even more.

## Contextual dimensions

The contextual dimensions highly depend on the context of the task at hand. In the case of relevancy dimension, the Semantic Publishing Challenge did not perform any relevant evaluation. Nevertheless, in this paper the coverage metric is addressed. To be more precise, in the 'Coverage' section, the coverage is thoroughly discussed. The Semantic Publishing Challenge does contribute to the timeliness dimension. To be more precise, thanks to its continuity, it is assured that at least every year the challenge is organized, a new dataset for the underlying CEUR-WS.org data is generated, boosting the freshness metric. In particular the final extraction has to be made from the evaluation dataset published a few days before the final submission deadline. As a conclusion, the challenge succeeded in indirectly promoting the coverage and timeliness dimensions; however, there is potential for other dimensions to be covered as well.

## Representational dimension

The representational dimension captures aspects related to the data design (*Zaveri et al., 2016*). As far as the interoperability dimension is concerned, the Semantic Publishing Challenge promotes the reuse of existing terms and vocabularies and, as shown in Table 4 and discussed in the 'Annotations' section, the Semantic Publishing Challenge achieves its goal of promoting the re-use of existing vocabularies, even though the corresponding metric is not evaluated automatically. Moreover, thanks to Task 3, the Semantic Publishing Challenge also promotes the re-use of existing terms. Even though it failed to attract participation, it is proven that such a task contributes into increasing the overall dataset quality. Thus, the challenge enables the produced datasets to cover even the representational quality dimension.

## CONCLUSIONS

One of the objectives of the Semantic Publishing Challenge is to produce Linked Data that contributes to improving scholarly communication. Nevertheless, the lessons learned from organizing this challenge are not only applicable in the case of a challenge on Semantic Publishing but in the case of other challenges too. Therefore, this work shed light not only on the three editions of this challenge organized by ourselves and distilled lessons learned from our experience, but we have also validated them against other challenges and concluded on general best practices for organizing such challenges. In a nutshell, continuity both in terms of the dataset and in terms of the tasks is important. Nevertheless, tasks should remain distinct, but they should refer to the same training and evaluation dataset, while participants' feedback should be taken into consideration to define or refine the tasks. Regarding the output, the larger the evaluation dataset is and the less overlapping with the training dataset, the best it is for verifying high coverage. The sooner the evaluation

tool is made available, the better it is for the quality of the final output. Finally, it is a critical incentive for the participants to know how their output is intended to be reused.

Besides the challenge's organizational aspects, we looked for evidence from the solutions proposed by the participants. Therefore, we analyzed them, reported our observations and came up with different conclusions related to Linked Data publishing practices followed by different participants. There are several positive aspects, among them the high participation and the quality of the produced results. This work allowed us to share those observations on semantifying scholarly data, using different ontological models, refining and extending existing datasets. Even though the Semantic Publishing Challenge focuses on scholarly data, the conclusions we draw based on our analysis are of interest for the entire community that publishes Linked Data. The possibility of sharing knowledge and solutions among participants was another key factor of the Semantic Publishing Challenge. In a nutshell, most solutions relied on generic and open-sourced tools, which allows and enables their reuse for corresponding cases. Solutions, and thus the tools that produce them, have improved from one edition to the other. Even though different methodologies were followed, there are certain prevailing approaches—based on structure/layout or on linguistics—which were instantiated in different ways. Despite the fact that tools diverge, the produced data model and final annotations converge, as solutions become more mature from one edition to the other, while well-known vocabularies are reused.

Last, we assessed how the challenge's organization reflects on the submitted solutions' output, namely how the challenge's organization affects the datasets' quality. We showed that indeed the challenge's organization may have a positive impact on increasing the quality of the Linked Data produced.

### Funding

Research activities described in this paper were funded by Ghent University, iMinds, the Institute for the Promotion of Innovation by Science and Technology in Flanders (IWT), the Research Foundation—Flanders (FWO) and the European Union under grant agreement no. 643410 (OpenAIRE2020) and others. The funders had no role in study design, data collection and analysis, decision to publish, or preparation of the manuscript.

### Grant Disclosures

The following grant information was disclosed by the authors:
Ghent University, imec.
Flanders Innovation & Entrepreneurship (VLAIO).
European Union: 643410.

### Competing Interests

The authors declare there are no competing interests.
Christoph Lange is an employee of Enterprise Information Systems.
Anastasia Dimou and Erik Mannens are employees of imec.

## Author Contributions

- Anastasia Dimou, Sahar Vahdati, Angelo Di Iorio and Christoph Lange conceived and designed the experiments, performed the experiments, analyzed the data, contributed reagents/materials/analysis tools, wrote the paper, prepared figures and/or tables, performed the computation work, reviewed drafts of the paper.
- Ruben Verborgh wrote the paper, prepared figures and/or tables, reviewed drafts of the paper.
- Erik Mannens reviewed drafts of the paper.

## Data Availability

    (1) SemPub2016: https://github.com/ceurws/lod/wiki/SemPub2016

    (2) SemPub2015: https://github.com/ceurws/lod/wiki/SemPub2015

    (3) SemPub2014: http://challenges.2014.eswc-conferences.org/index.php/SemPub.

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
