# Peer review of "Challenges as enablers for high quality Linked Data: insights from the Semantic Publishing Challenge"

_PeerJ Computer Science, doi:10.7717/peerj-cs.105_

## Round 0.1 · original submission · Major Revisions

· Academic Editor

Major Revisions

The reviewers felt that the work is interesting, but have indicated that major revisions are needed. We encourage you to revise the manuscript and resubmit it for reconsideration. If you decide to submit a revised version you must address all the reviewers' comments. We look forward to seeing your revision.

Reviewer 1 ·

Basic reporting

The paper is written in clear, professional English. The structure of the paper is clear and it corresponds to the presented topic although it does not fully conform to the recommended PeerJ standard. The Introduction and Background sections provide the necessary context, the referenced sources are relevant to the topic. The presented tables provide important and useful additional information. The table formatting should be improved, for example the structure of Table 2 is not very clear and it takes time to understand the meaning of the rows and overlapping columns. Raw datasets mentioned in the paper seem to be available in a public repository at GitHub.com.

Experimental design

The paper focuses on summarizing the experience gained during three years of organizing The Semantic Publishing Challenge. Thus, it is not a typical research paper presenting hard experimental results. The relevant methods of linked dataset creation are mentioned only briefly and the main focus is given to sharing the experience and "lessons learned". In this part, the summarization of the findings corresponds to the evolution of the Challenge - authors thoroughly describe the aspects of the Challenge that turned out to be beneficial or counterproductive. The description is sufficiently detailed. The only thing I am missing is a more thorough description of the process of preparing the "golden standard" data used for evaluating the submitted datasets and evaluation process in general. Many questions arise here: Which tools have been used for creating the "golden standard"? Was the result validated with the source documents (how?) or was any other data source used for its validation? Does some of the submitted tools actually provide better results in some situation that the golden standard and could such situation be detected properly?

Validity of the findings

The findings regarding the Challenge organization are compared and sufficiently validated with the results of a survey conducted among the organizers of similar challenges. Moreover, the paper provides an interesting statistical summary of the properties of the obtained data sets including the usage of the existing ontologies, their concepts, etc. which come from the available raw data.

Additional comments

Although the topic of the paper is not typical for informatics and computer science, it is definitely very interesting and the experience is worth sharing. I find the paper well structured and the findings are presented in detail. I recommend only minor revisions regarding (1) the table data presentation and (2) more detailed description of the Challenge evaluation process and the "golden standard" preparation.

Reviewer 2 ·

Basic reporting

"No Comments"

Experimental design

"No Comments"

Validity of the findings

"No Comments"

Additional comments

The paper provides a description and analysis of the Semantic Web Publishing Challenge, deriving a set of challenge organization best practices and providing a description of Linked Data produced as a result of the challenge. The proposed analysis is relevant and the focussed scope of the analysis corpus (as well as some of the author's involvement with the challenge organization) brings the opportunity for a deeper analysis of the problem space. The paper needs more effort on the analysis and on communication side. Improvement points are provided below.

- As it is the paper is rather descriptive and fails to create a process of abstraction and generalization of interesting patterns within the challenge. This is the main limitation of the paper. To address this, the paper needs to be seriously reworked. For example the tables in the paper concentrate on comparing lower-level attributes, rather then using higher level categories which can shed some light on the challenges and insights of Linked Data production and publication.

- The authors here have a great opportunity of providing the core insights, from a 'lab' perspective (the challenge here working as an experiment) on the critical aspects of representation, infrastructure and cultural that influence the practice of Linked Data publication. Can you dig into the dataset heterogeneity problems, for example? Which aspects of the representation provide more complete/high quality Linked Data?

- The paper is well-written but it lacks structure. As it is, the paper is a collection of low-level descriptions of observations and attributes of the challenges put together as a continuous text flow. The paper fails to generalize the observations into higher-level categories. Introducing more structure into the paper (Bullet points, subsections) is a very central improvement for the discourse.

- Some more 'soft' but very fundamental aspects are left out of the analysis at Section 3. In which communities the Semantic Web Publishing challenge had better traction (Semantic Web, Biomedical domain)? How the perception about the core communities which would be the main stakeholders in the challenge changed?

- Comparative analysis with related work is limited (and for a good reason as there are not many works on the evaluation of campaigns). One suggestion would be to better link Section 4 with related work on Linked Data Quality? Information retrieval has a long tradition on evaluation campaigns. Can you link to any work on this community?

External reviews were received for this submission. These reviews were used by the Editor when they made their decision, and can be downloaded below.

---

## Round 0.2 · accepted · Accept

· Academic Editor

Accept

The authors have addressed all the reviewers' comments in this new submission.

External reviews were received for this submission. These reviews were used by the Editor when they made their decision, and can be downloaded below.